🔓 | **Open Peer Review** | Environmental Microbiology | Research Article

# Screening microbial inhibitors of *Pseudogymnoascus destructans* in Northern China

Shaopeng Sun,[1] Mingqi Shan,[1] Zihao Huang,[1] Yihan Lv,[1] Zizhen Wei,[1] Mingqi Shen,[1] Keping Sun,[2] Zhongle Li,[1,3] Jiang Feng[1,3]

**ABSTRACT** White-nose syndrome is caused by *Pseudogymnoascus destructans,* leading to the near extinction of multiple bat populations in North America. This fungal pathogen has also been detected in China, but the prevalence and loads are relatively low in the hosts and environment. Previous studies have screened bat skin microbiomes in China to identify microbes that inhibit the growth of *P. destructans*. However, there is limited information on bacterial genera that possess properties that inhibit *P. destructans* in bat cave environments in China, particularly regarding antifungal metabolic pathways. We isolated 29 bacterial strains that have the ability to inhibit growth of *P. destructans* from the skin of bats and soil samples in China. These strains primarily belonged to several genera, including *Acinetobacter*, *Pseudomonas*, and *Serratia*. Gas chromatography-mass spectrometry analysis identified volatile organic compounds from strains that inhibit *P. destructans*, showing that 100 µL of α-Pinene, 2-Undecanone, 2-Nonanone, 2,5-Dimethylpyrazine, as well as 10 µL Benzaldehyde and Thujone, completely inhibited the growth of *P. destructans* and caused morphological damage to the mycelium. The soluble secondary metabolites from the antagonistic strains indicated that the bioactive compounds were predominantly small-molecule organic substances. Whole-genome sequencing of these antagonistic strains revealed that the most enriched potential antifungal gene clusters were associated with bacteriocins, siderophores, and β-lactones. β-Lactones were the primary gene cluster against *P. destructans*, and chitinases may play a crucial role in the antifungal process.

**IMPORTANCE** Bat skin and environmental microbiota may influence the colonization and persistence of *Pseudogymnoascus destructans*, thereby potentially affecting the occurrence of white-nose syndrome. Examining differences in these gene clusters contributes to understanding variation in the capacity of bacterial groups to have characteristics that inhibit *P. destructans*. This study lays the foundation for further exploration and elucidation of the mechanisms by which bacteria from bat skin and roosting environments suppress *P. destructans* growth *in vitro*.

**KEYWORDS** *Pseudogymnoascus destructans*, bat, bacteria, volatile organic compound, biosynthetic gene cluster

The threat of fungal pathogens in wildlife populations is escalating. Examples include the widespread emergence of *Batrachochytrium dendrobatidis*, a chytrid fungus which causes chytridiomycosis in amphibians (1), and *Pseudogymnoascus destructans*, an ascomycete that is the etiological agent of white-nose syndrome in North American hibernating bats (2). Both pathogenic fungi attack the skin of the host, and host-associated skin microbiota can be antagonistic by secreting antifungal compounds or competing for resources. One possible biocontrol strategy to reduce or inhibit risks of these, and other fungal infection is through the use of host-associated probiotic bacteria (3, 4). Moreover, because these fungal pathogens can persist outside the host

Address correspondence to Zhongle Li, lzy1514316@126.com, or Jiang Feng, fengj@nenu.edu.cn.

The authors declare no conflict of interest.

See the funding table on p. 15.

for extended periods (5), a potential mitigation approach can be to minimize their environmental loads (6), for example, via the application of environmental probiotics.

White-nose syndrome (WNS) was first identified in 2006 and has caused significant mortality in bat populations (2, 7), due to the colonization of *P. destructans* on the mouths, noses, and wing membranes of hibernating bats, leading to skin tissue damage and premature depletion of stored fat (8). Compared to North America, where this fungal pathogen can have dramatic impacts on bat populations, there are relatively low *P. destructans* loads in the environment, and in bats, in China (9). Indeed, several microbial inhibitors of *P. destructans* have been identified from the skin of bats in China (e.g., *Pseudomonas*, *Rhodococcus*, and *Acinetobacter*) (10). This could result from the secretion of phenazine-1-carboxylic acid and various volatile organic compounds (VOCs) that suppress *P. destructans* growth *in vitro* (11). However, research on bacterial genera with anti-*P. destructans* properties in bats and their surrounding cave environments is limited, and the active components of their metabolites require urgent investigation.

Despite numerous studies elucidating the probiotic properties of bacterial strains that inhibit *P. destructans in vitro* (12, 13), little is known about the diversity of antifungal metabolic pathways or the specific gene clusters that contribute to the effects of different bacterial taxa on *P. destructans*. With the rapid development of whole-genome sequencing technologies, extracting and analyzing genetic information from microorganisms has become increasingly efficient (14). Prokaryotic genomes have garnered significant attention due to their potential to encode bioactive secondary metabolites, offering promising avenues for the discovery of antifungal compounds. For example, compounds such as prodigiosin, amphotericin B, and griseofulvin possess antifungal

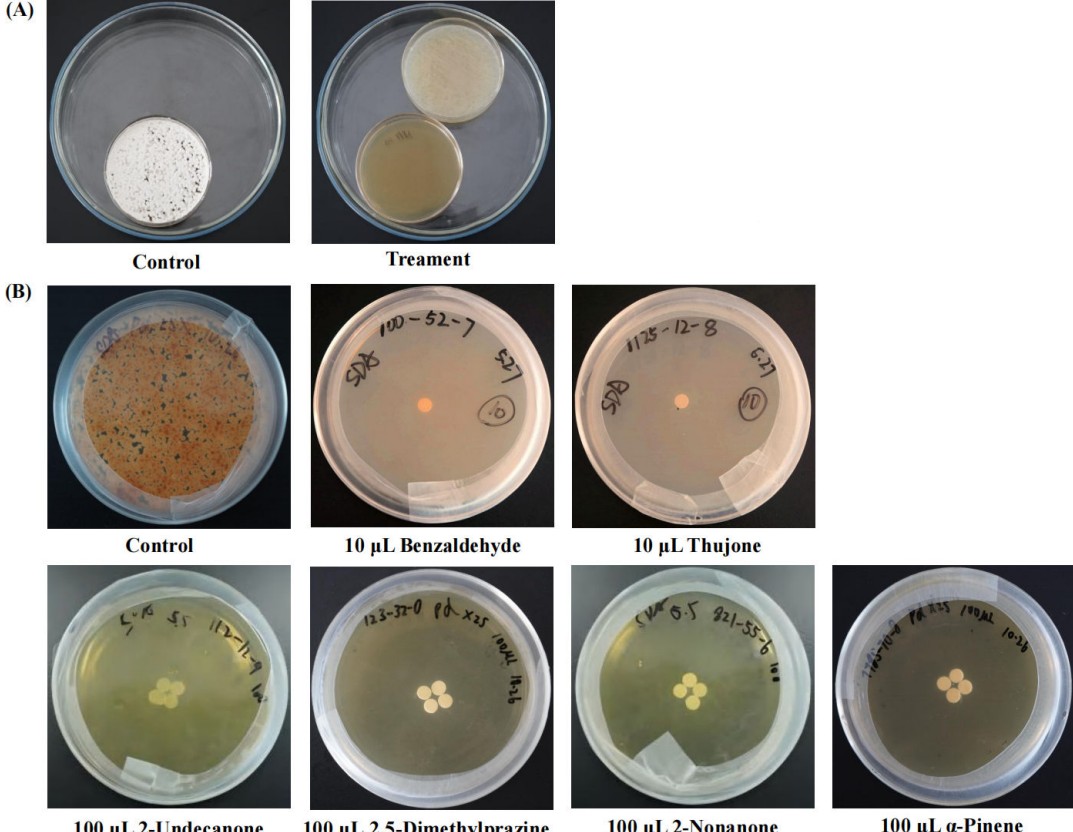

**FIG 1** Evaluation and validation of the effects of volatile organic compounds (VOCs) from antagonistic strains against *P. destructans*. (A) Co-cultivation of antagonistic strains with *P. destructans* for 14 days. Control: *P. destructans* alone. (B) Co-cultivation of VOCS with *P. destructans* for 14 days. Control: *P. destructans* alone. Images show the agar surface (A) and reverse side (B) of the plates.

activity (15–17). In recent years, bioinformatics tools such as antiSMASH have been widely applied in prokaryotic genome analysis to predict and identify biosynthetic gene clusters (BGCs) associated with secondary metabolism (18). Potential antifungal-active BGCs include bacteriocins, siderophores, and other classes of metabolites (19, 20), providing an effective approach for screening novel antifungal molecules.

In this study, we isolated, identified, and analyzed bacteria with properties that can inhibit the growth of *P. destructans* from skin samples of hibernating bats and soil samples from roosting caves in northeastern China. Through *in vitro* assays, we screened strains that effectively inhibit *P. destructans* growth. We then selected 29 antagonistic strains for whole-genome sequencing to identify the gene clusters associated with the antimicrobial mechanisms. We also explored the VOCs and small molecules produced by the antagonistic strains to assess them for their inhibitory effects on *P. destructans*. This research provides scientific evidence supporting the potential development of microbe-based biological control agents against *P. destructans*, offering novel insights into the mitigation of WNS in bats.

## RESULTS

### Bat skin and soil microbiome

From the 50 bat skin samples, a total of 43 bacterial species were isolated and identified, with the most prominent genera being members of *Bacillus*, *Acinetobacter*, *Pseudomonas*, and *Staphylococcus*. In addition, from the 50 soil samples, 32 bacterial species were isolated, primarily members of *Acinetobacter*, *Bacillus*, and *Pseudomonas*.

### Analysis of agar plate challenge assay

A total of nine bacterial strains isolated from bat skin completely inhibited the growth of *P. destructans*. These included members of *Pantoea*, *Pseudomonas*, *Acinetobacter,* and *Serratia*. In addition, 10 bacterial strains isolated from soil samples completely inhibited the growth of *P. destructans*, within the *Acinetobacter*, *Pseudomonas,* and *Stenotrophomonas* (Fig. 1A; Table 1).

**TABLE 1**  Antagonistic strains against *P. destructans* in the agar plate challenge assay

| Sampling sites | Sources | Strains | Inhibition effect |
|---|---|---|---|
| Di Cave | Skin (*Rhinolophus ferrumequinum*) | *Acinetobacter*_1 | Complete |
| Di Cave | Skin (*Rhinolophus ferrumequinum*) | *Acinetobacter*_2 | Complete |
| Di Cave | Skin (*Rhinolophus ferrumequinum*) | *Acinetobacter*_18 | Complete |
| Bat Cave | Skin (*Myotis ricketti*) | *Pantoea*_1 | Complete |
| New Cave | Skin (*Myotis petax*) | *Acinetobacter*_4 | Complete |
| Di Cave | Skin (*Rhinolophus ferrumequinum*) | *Acinetobacter*_16 | Complete |
| Di Cave | Skin (*Murina leucogaster*) | *Acinetobacter*_17 | Complete |
| Temple Cave | Skin (*Murina leucogaster*) | *Pseudomonas*_1 | Complete |
| New Cave | Skin (*Rhinolophus ferrumequinum*) | *Serratia* _1 | Complete |
| New Cave | Soil | *Acinetobacter*_11 | Complete |
| New Cave | Soil | *Pseudomonas*_4 | Complete |
| New Cave | Soil | *Acinetobacter*_19 | Complete |
| Temple Cave | Soil | *Acinetobacter*_9 | Complete |
| Di Cave | Soil | *Acinetobacter*_10 | Complete |
| Temple Cave | Soil | *Acinetobacter*_12 | Complete |
| Gezi Cave | Soil | *Acinetobacter*_6 | Complete |
| Gezi Cave | Soil | *Stenotrophomonas*_1 | Complete |
| Gezi Cave | Soil | *Pseudomonas*_3 | Complete |
| Gezi Cave | Soil | *Pseudomonas*_2 | Complete |

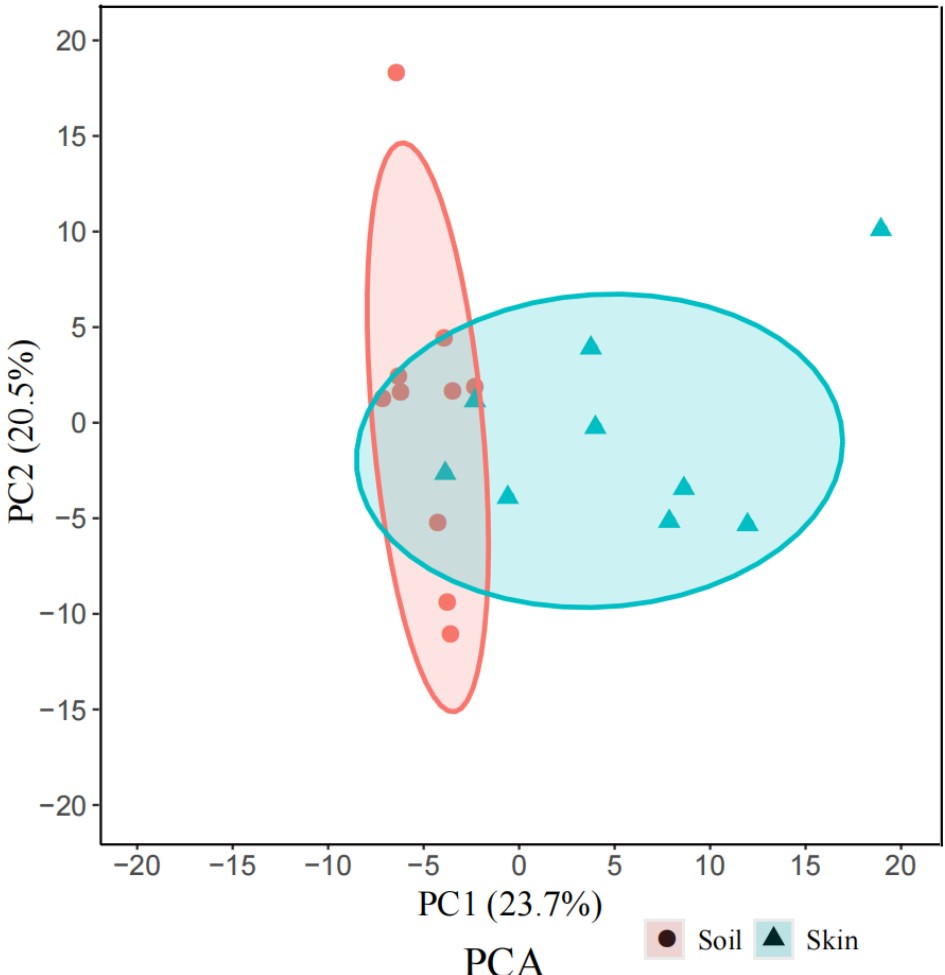

**FIG 2** Principal component analysis of volatile organic compounds (>1%) produced by antagonistic strains against *P. destructans* from bat skin and soil samples.

## Identification and verification of volatile organic compounds produced by antagonistic strains against *P. destructans*

Volatile organic compounds (VOCs) produced by bacterial strains that inhibited *P. destructans* growth through contact-independent mechanisms were identified using gas chromatography–mass spectrometry (GC-MS). Permutational multivariate analysis of variance (PERMANOVA) based on the Bray-Curtis similarity matrix indicated a significant difference in the relative abundances of volatile organic compounds (>1%) between groups (Pseudo-$F_{2, 19}$ = 2.67, $R^2$ =0.14, $P$ = 0.002; Fig. 2). The major VOC components were 2-Undecanone (CAS:112-12-9), 2-Nonanone (CAS:821-55-6), 2-Tridecanone (CAS:593-08-8), 1-Undecene (CAS:821-95-4), Benzaldehyde (CAS:100-52-7), and Phenylethanol (CAS:60-12-8) (Fig. 3; Table S1). When exposed to 100 µL of α-pinene, 2-Undecanone, 2,5-Dimethylpyrazine, 2-Nonanone, the growth of *P. destructans* was inhibited; likewise growth was inhibited by 10 µL of Benzaldehyde and Thujone (Fig. 1B).

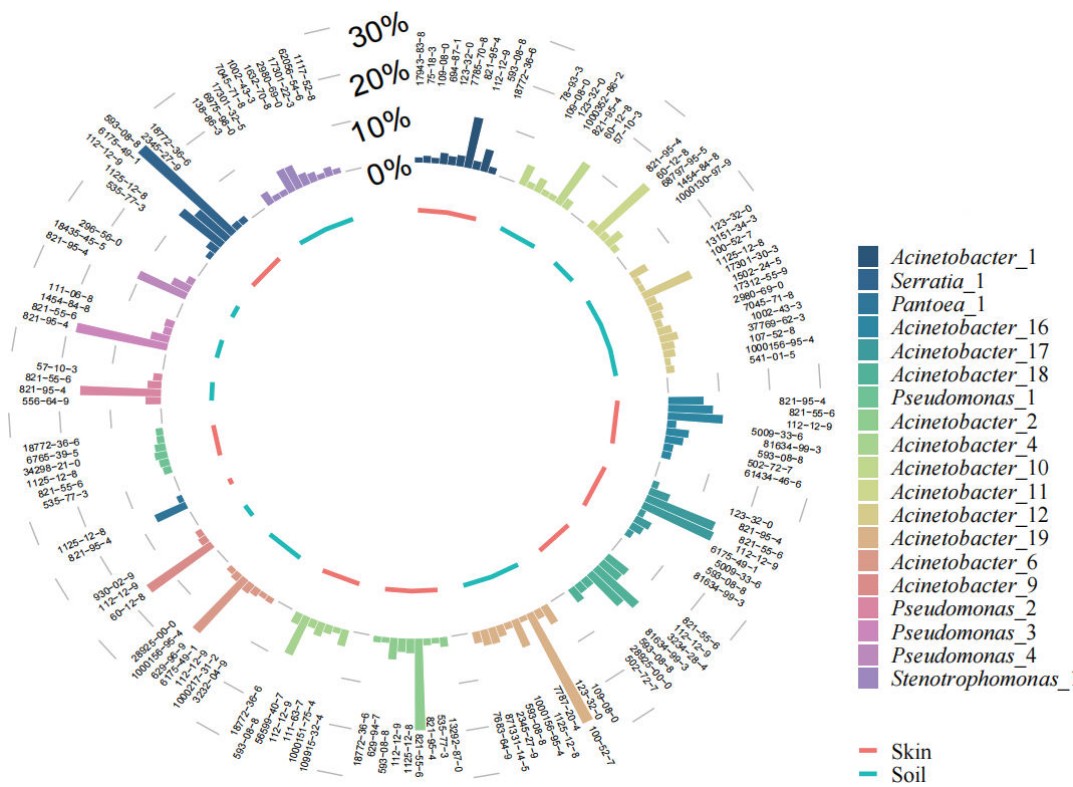

**FIG 3** Types and relative abundances (>1%) of volatile organic compounds produced by antagonistic strains against *P. destructans* from bat skin and soil.

## Effect of volatile organic compounds on the morphology of *P. destructans* mycelium

Scanning electron microscopy revealed visible morphological changes in *P. destructans* mycelia 14 days after exposure to VOCs. Mycelia were deformed, shortened, and lacked fullness in the treatment group but were smooth, slender, uniformly thick, and morphologically intact in the control group (Fig. 4).

## Analysis of cell-free supernatant challenge assays

Among the bacterial isolates from bat skin samples, 11 strains were inhibited by more than 50%, including *Acinetobacter*, *Bacillus,* and *Pantoea*. In soil samples, there were six bacterial strains with inhibition rates exceeding 50%, all within the *Acinetobacter* (Table 2).

## Preliminary analysis of protein fractions and molecular weight range and polarity in the fermentation broth of antagonistic strains

In the cell-free supernatant challenge assays, the active components of 17 antagonistic strains with an inhibition rate greater than 50% were investigated. Protein fractions had minimal inhibitory activity against *P. destructans*, whereas non-protein components had strong inhibitory effects (Table S2).

After dialysis of the fermentation broths from 17 antagonistic strains, the dialysates exhibited strong inhibitory activity against *P. destructans*, while the retentates showed no inhibitory effect. These results suggest that the active compounds produced by all 17 strains have a molecular weight below 1,000 Da, indicating that they are small-molecule compounds (Table S2).

The fermentation broths of 17 antagonistic strains were extracted with different organic solvents, revealing variations in inhibitory activity against *P. destructans*. Among these, four strains showed the highest inhibition in 1-butanol extracts, five in petroleum

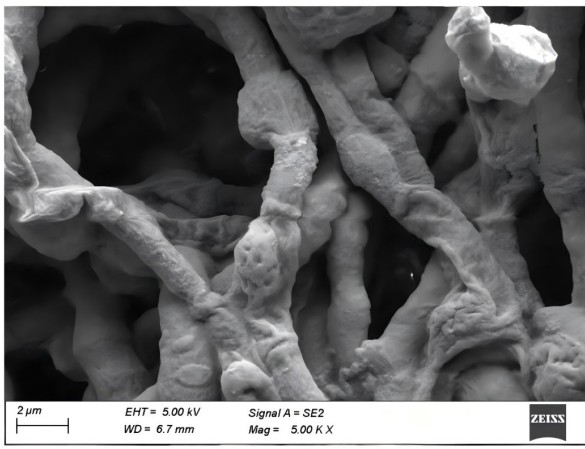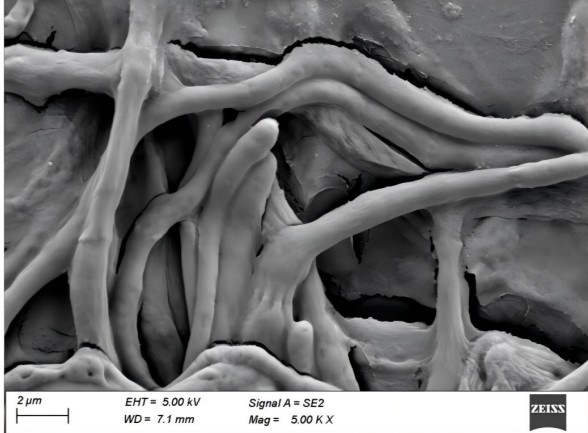

FIG 4 Scanning electron microscopy observation of the effects of volatile organic compounds (VOCs) on *P. destructans* mycelial morphology. Left: Treatment group where *P. destructans* mycelia were exposed to VOCs. Right: Control group with only *P. destructans* mycelia. Magnification: 5,000×; Scale bar: 2 µm.

ether extracts, and seven in ethyl acetate extracts. Notably, one strain exhibited the highest inhibition rate in the aqueous phase (Table 3).

## Genomic analysis of antagonistic strains

A total of 28 samples (14 skin and 14 soil) were subjected to whole-genome sequencing. The number of contigs per genome ranged from 5 to 961, with N50 contig lengths ranging from 20,808 to 927,625 bp. Genome sizes ranged from 1,476,802 to 12,148,644 bp, and the GC content ranged from 31.73% and 67.58%, filtering out low-quality assemblies. Biosynthetic gene clusters (BGCs) were predicted using AntiSMASH, identifying 23 distinct types, including NRPS, bacteriocin, siderophore, betalactone, and LAP (Table S3). NRPS was the most frequently detected BGC, present in 28 of the isolates. PERMANOVA based on the Bray-Curtis similarity matrix indicated that there was no significant difference in BGC types and abundances between skin and soil samples (Pseudo-$F_{2, 28}$= 2.23, $R^2$= 0.08, $P$ = 0.055; Fig. S1). Due to the limited number of shared genes at the taxonomic level, comparative analyses of common genes within the BGCs were conducted at the genus level. Within *Acinetobacter*, 11 strains shared a siderophore (8 shared genes), 12 strains shared a bacteriocin (8 shared genes), and 11 strains harbored a beta-lactone cluster related to the antifungal compound fengycin (7 shared genes) (Fig. 5). In addition, a LAP cluster was predicted in *Acinetobacter*_4 (Table S3). Chitinase genes were also predicted in *Acinetobacter*_4 and *Acinetobacter*_19 (Table S4).

In *Bacillus*, two isolates shared a siderophore (12 shared genes) with an average nucleotide identity of 99%, a terpene (23 shared genes) with 99.36% identity, and a bacteriocin (five shared genes) with 98.9% identity (Fig. S2). Additionally, chitinase genes were predicted in two isolates, sharing two homologous genes (Table S4). In *Bacillus*_2, NRPS-T1PKS gene cluster was predicted, showing 100% sequence homology to the zwittermicin A cluster from *Bacillus cereus* (Tables S3 and S5). In the four *Pseudomonas* isolates, no shared siderophores, bacteriocins, or terpenes were found. However, *Pseudomonas*_1 had a chitinase cluster with a high inhibition rate, as did the *Serratia*_1 isolate (Table S4). In *Pantoea*_1, a terpene with 100% homology to carotenoid was predicted (Table S5). An uncharacterized type III polyketide synthase (T3PKS) was identified in *Mammaliicoccus*_1 and *Staphylococcus*_1 (Table S3). In the antagonistic strains, key genes involved in the synthesis of VOCs—including 2,5-Dimethylpyrazine and Benzaldehyde—were detected in *Acinetobacter*_1, *Acinetobacter*_10, *Acinetobacter*_12, and *Acinetobacter*_19 (Table 4).

**TABLE 2** Inhibition rates of antagonistic strains

| Sampling sites | Sources | Strains | Inhibition rates |
|---|---|---|---|
| Di Cave | Skin (*Rhinolophus ferrumequinum*) | *Acinetobacter*_1 | 56.78% |
| Di Cave | Skin (*Rhinolophus ferrumequinum*) | *Acinetobacter*_2 | 53.61% |
| New Cave | Skin (*Myotis petax*) | *Mammaliicoccus*_1 | 55.54% |
| Di Cave | Skin (*Rhinolophus ferrumequinum*) | *Acinetobacter*_3 | 56.69% |
| Bat Cave | Skin (*Myotis ricketti*) | *Staphylococcus*_1 | 77.32% |
| Bat Cave | Skin (*Myotis ricketti*) | *Bacillus*_1 | 59.96% |
| New Cave | Skin (*Myotis petax*) | *Acinetobacter*_4 | 68.59% |
| Bat Cave | Skin (*Myotis ricketti*) | *Pantoea*_1 | 63.23% |
| New Cave | Skin (*Rhinolophus ferrumequinum*) | *Serratia*_1 | 64.22% |
| Bat Cave | Skin (*Rhinolophus ferrumequinum*) | *Bacillus*_2 | 60.60% |
| Temple Cave | Skin (*Murina leucogaster*) | *Pseudomonas*_1 | 40.35% |
| Bat Cave | Skin (*Rhinolophus ferrumequinum*) | *Acinetobacter*_5 | 58.42% |
| Gezi Cave | Soil | *Acinetobacter*_6 | 54.90% |
| Gezi Cave | Soil | *Acinetobacter*_7 | 61.75% |
| Temple Cave | Soil | *Acinetobacter*_8 | 55.96% |
| Temple Cave | Soil | *Acinetobacter*_9 | 40.25% |
| Di Cave | Soil | *Acinetobacter*_10 | 42.25% |
| Gezi Cave | Soil | *Pseudomonas*_2 | 30.52% |
| New Cave | Soil | *Acinetobacter*_11 | 44.35% |
| Gezi Cave | Soil | *Pseudomonas*_3 | 32.21% |
| Temple Cave | Soil | *Acinetobacter*_12 | 51.92% |
| Di Cave | Soil | *Acinetobacter*_13 | 54.18% |
| New Cave | Soil | *Pseudomonas*_4 | 21.21% |
| Di Cave | Soil | *Acinetobacter*_15 | 60.60% |
| Di Cave | Skin (*Rhinolophus ferrumequinum*) | *Acinetobacter*_16 | 43.69% |
| Di Cave | Skin (*Murina leucogaster*) | *Acinetobacter*_17 | 42.30% |
| Di Cave | Skin (*Rhinolophus ferrumequinum*) | *Acinetobacter*_18 | 30.30% |
| Gezi Cave | Soil | *Stenotrophomonas*_1 | 47.33% |
| New Cave | Soil | *Acinetobacter*_19 | 48.56% |

## DISCUSSION

In this study, we isolated anti-*P. destructans* bacterial strains from bat skin and soil from bat caves. Inhibitory effects of these strains occurred both through contact and non-contact modes; non-contact inhibition was attributed to the action of volatile organic compounds (VOCs). The dominant antagonistic genera included *Pantoea*, *Pseudomonas*, *Acinetobacter*, *Serratia*, and *Stenotrophomonas* (Table 1). Among them, *Acinetobacter* was the most prevalent and is known to be widely distributed in natural environments and on bat skin (21–23). Indeed, *Acinetobacter* is known to have antifungal properties, for example, inhibiting pathogenic *Stemphylium lycopersici* on tomatoes via VOCs (24). Antifungal activity through VOCs has also been reported in isolates from other bacterial genera (25–28). There were significant differences in the VOCs produced by bat skin and soil microbiota (Fig. 2), likely as a result of variations in the bacterial composition in these distinct environments.

Several VOCs have been tested for their for their inhibitory effects against *P. destructans*, including 2-Undecanone and 2-Nonanone (10). In this study, these

**TABLE 3** Polarity analysis of antagonistic strains

| Strains | 1-Butanol | Petroleum ether | Ethyl acetate | Dichloromethane | Aqueous phase |
|---|---|---|---|---|---|
| *Acinetobacter*_1 | 70.43% | 95.2% | 90.01% | 0.8% | 23.21% |
| *Acinetobacter*_2 | 0.32% | 81.02% | 89.63% | 80.82% | 67.36% |
| *Mammaliicoccus*_1 | 79.18% | 90.36% | 93.74% | 26.75% | 57.47% |
| *Acinetobacter*_3 | 75.24% | 49.51% | 1.2% | 23.79% | 72.41% |
| *Staphylococcus*_1 | 63.03% | 76.53% | 36.55% | 0.2% | 48.78% |
| *Bacillus*_1 | 23.07% | 73.78% | 67.52% | 0% | 58.97% |
| *Acinetobacter*_4 | 70.17% | 23.39% | 0% | 17.89% | 42.48% |
| *Pantoea*_1 | 51.81% | 74.58% | 83.01% | 63.60% | 59.30% |
| *Serratia*_1 | 0% | 90.33% | 0% | 0% | 20.36% |
| *Bacillus*_2 | 99.88% | 83.62% | 93.74% | 87.77% | 28.29% |
| *Acinetobacter*_5 | 0% | 54.52% | 53.98% | 24.73% | 76.32% |
| *Acinetobacter*_6 | 63.52% | 0% | 4.32% | 28.61% | 38.03% |
| *Acinetobacter*_7 | 0% | 0% | 81.09% | 0% | 48.36% |
| *Acinetobacter*_8 | 0% | 73.79% | 91.72% | 21.46% | 12.21% |
| *Acinetobacter*_12 | 20.31% | 56.21% | 21.11% | 0% | 46.32% |
| *Acinetobacter*_13 | 0% | 0% | 48.88% | 0% | 33.51% |
| *Acinetobacter*_15 | 0% | 50.12% | 67.61% | 40.6% | 43.5% |

inhibitory VOCs were detected in several bacterial strains (Fig. 1A), as well as several other inhibitory compounds, including α-Pinene, Thujone, 2,5-Dimethylpyrazine, and Benzaldehyde (Fig. 1B). The antimicrobial properties of α-Pinene have primarily been observed in plants (29, 30), but has also been detected in *Burkholderia tropica*, a bacteria that possesses antifungal activity (31). Thujone is a primary VOC in plant essential oils with antifungal activity (32, 33). 2,5-Dimethylpyrazine is a VOC is that is widespread in soils with antifungal properties (34), and has been observed to be produced by *Lysobacter capsici* AZ78, which showed inhibitory effects against soil-borne plant pathogens (35). This provides theoretical support for its potential in reducing the *P. destructans* load in the environment. We identified several genes associated with the synthesis of 2,5-Dimethylpyrazine (Table 4), which is formed through a non-enzymatic reaction involving the spontaneous dimerization and oxidative dehydration of aminoacetone. Additionally, amino phenyl ketone, a precursor in this pathway, can be synthesized by TDH using L-threonine or by ydfG using L-isoleucine as substrates (36). Benzaldehyde and its derivatives exhibit strong antimicrobial activity against *Aspergillus fumigatus*, *A. flavus*, *A. terreus*, and *Penicillium* by disrupting their redox systems (37, 38). Relevant synthetic enzymes were identified through KEGG (Table 4). These volatile organic compounds have low toxicity, rapid reaction, potent antifungal activity, and can diffuse via aerosols, indicating potential application value. However, their practical use in natural environments still requires further validation (39). Scanning electron microscopy further revealed that VOCs caused significant morphological changes in *P. destructans* mycelia, indicating severe damage to the mycelial structure (Fig. 4).

The contact inhibition of *P. destructans* is primarily attributed to the soluble secondary metabolites of the antagonistic strains. Strains from *Acinetobacter*, *Bacillus*, *Mammaliicoccus*, *Pantoea*, *Staphylococcus*, and *Serratia* showed high inhibition rates (Table 2). Indeed, *Mammaliicoccus* and *Staphylococcus* have been shown to use bacteriocins to inhibit pathogenic fungi (40). Likewise, *Acinetobacter* production of erythromycin can inhibit the growth of several plant pathogens (41), isolates of *Pantoea* have antifungal activity against *Ceratocystis fimbriata* (15), and *Serratia* production of prodigiosin exhibited inhibitory effects on both bacteria and fungi (42, 43). By comparing the inhibitory effects of protein and supernatant on *P. destructans*, we found that the non-protein fractions exhibited inhibitory activity (Table S2), further confirming that it consisted of small molecules (i.e., molecular weight <1,000 Da) (Table S2). Organic solvent extraction indicated that these small molecules were organic (Table 3). Because secondary metabolites are low molecular weight organic compounds synthesized

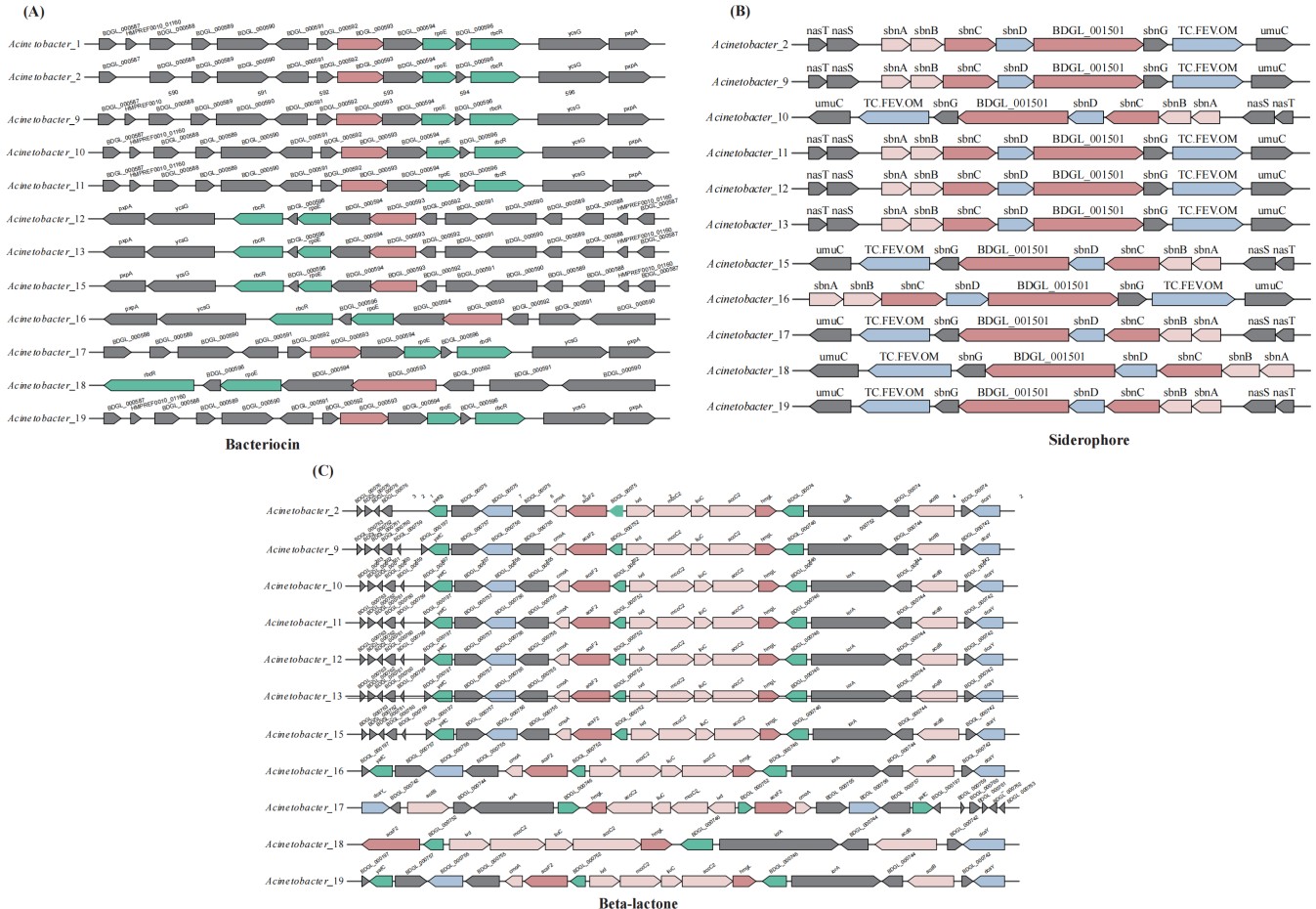

**FIG 5** Comparison of biosynthetic gene clusters in *Acinetobacter*. Isolates are labeled by genus and ID number. Gene functions are color-coded: dark red represents core biosynthetic genes; pink represents additional biosynthetic genes; blue represents transport-related genes; green represents regulatory genes; gray represents other genes. Panels show clusters predicted to encode (A) Bacteriocin (N = 12), (B) Siderophore (N = 11), and (C) β-lactone (N = 11).

by microorganisms in the later stages of growth (44), this confirms that secondary metabolites are responsible for the observed inhibitory effects.

After whole-genome sequencing of 28 strains that had inhibitory effects on *P. destructans*, 23 biosynthetic gene clusters were identified. Genome structures of these strains did not differ between those found in skin and soil samples (Fig. S1), possibly as a result of environmental adaptation (45). Because of the high diversity of gene clusters, here, we focus on three commonly present gene clusters that potentially contribute to antifungal activity. Bacteriocins are antimicrobial peptides (AMPs) synthesized by bacterial ribosomes that serve as natural defense mechanisms in microbial communities and can influence competition for resources with other microorganisms in the same environment (46). In our data set, 12 *Acinetobacter* isolates shared a bacteriocin comprising eight shared genes (Fig. 5A). Similarly, when comparing two strains with the largest difference in inhibition rates (60.60% and 30.21%), both shared eight genes and exhibited identical nucleotide sequences, suggesting that bacteriocins are unlikely to be the key factor in the inhibition of *P. destructans*. In two *Bacillus* strains with similar inhibition rates, while there were some gene differences in the shared bacteriocins (eight shared genes) (Fig. S2), these genes seem unlikely to be responsible for the variation in antifungal activity.

Iron plays an important role in several metabolic pathways, and when iron is limiting, microorganisms often produce siderophores that are small-molecule compounds to enhance iron uptake. As a consequence, bacterial siderophores can restrict

**TABLE 4** Key genes involved in the production of volatile organic compounds by antagonistic strains

| Volatile organic compounds | Strains | Chromosome location | Gene annotation | Gene_id |
|---|---|---|---|---|
| Benzaldehyde | *Acinetobacter*_19 | S30_GM003118 25984-27045 | Phenol 2-monooxygenase Domain-containing protein | dmpP, poxF, tomA5 |
| | | S30_GM003119 27060-27422 | Phenol hydroxylase subunit P4 | dmpO, poxE, tomA4 |
| | | S30_GM003120 27482-29008 | Aromatic/alkene/methane Monooxygenase Hydroxylase/oxygenase subunit alpha | dmpN, poxD, tomA3 |
| | | S30_GM003121 29054-29320 | MmoB/DmpM family protein | dmpM, poxC, tomA2 |
| | | S30_GM003122 29332-30333 | Aromatic/alkene monooxygenase | dmpL, poxB, tomA1 |
| | | S30_GM003123 30355-30642 | Phenol hydroxylase subunit | dmpK, poxA, tomA0 |
| | *Acinetobacter*_12 | S21_GM001984 86620-87681 | Phenol 2-monooxygenase Domain-containing protein | dmpP, poxF, tomA5 |
| | | S21_GM00198587696-88058 | Phenol hydroxylase subunit P4 | dmpO, poxE, tomA4 |
| | | S21_GM00198688118-89644 | Aromatic/alkene/methane Monooxygenase Hydroxylase/oxygenase subunit alpha | dmpN, poxD, tomA3 |
| | | S21_GM00198789690-89956 | MmoB/DmpM family protein | dmpM, poxC, tomA2 |
| | | S21_GM00198889968-90969 | Aromatic/alkene monooxygenase | dmpL, poxB, tomA1 |
| | | S21_GM00198990991-91278 | Phenol hydroxylase subunit | dmpK, poxA, tomA0 |
| 2,5-Dimethylpyrazine | *Acinetobacter*_12 | S21_GM001792113243-114001 | 3-Hydroxy acid dehydrogenase/Malonic semialdehyde reductase | ydfG |
| | *Acinetobacter*_19 | S30_GM000971105174-105932 | 3-Hydroxy acid dehydrogenase/Malonic semialdehyde reductase | ydfG |
| | *Acinetobacter*_10 | S17_GM0014225374-6132 | 3-Hydroxy acid dehydrogenase/Malonic semialdehyde reductase | ydfG |
| | *Acinetobacter*_1 | S1_GM00203655491-56519 | L-threonine 3-dehydrogenase | tdh |
| | | S1_GM002196105034-105792 | 3-Hydroxy acid dehydrogenase/Malonic semialdehyde reductase | ydfG |

the bioavailability of fungal iron (47, 48). In our data set, 11 *Acinetobacter* strains shared a siderophore with 8 common genes (Fig. 5B). To explore the potential role of these siderophores in the growth of *P. destructans*, we compared two strains with the most significant difference in inhibition rates (60.60% and 30.21%). These strains shared eight genes with identical sequences, suggesting that siderophores are not a key determinant of antifungal activity in *Acinetobacter*. Likewise, two *Bacillus* isolates with similar inhibition rates (59.96% and 60.60%), eight shared siderophore genes had 99% nucleotide homology, making it unlikely that siderophore genes directly explain variation in inhibition rates among strains (Fig. S2).

β-Lactones possess high ring strain, electrophilicity, and reactivity due to their four-membered heterocyclic structure that is found widely in microorganisms and exhibit antifungal properties (49). β-Lactones exert their effects as a result of several mechanisms (e.g., inactivating serine acetyltransferase), which prevents fungi from synthesizing methionine (50). When comparing *Acinetobacter* isolates, 11 strains shared a β-lactone (7 shared genes) (Fig. 5C). Although *Acinetobacter*_28 (30.21%) also produced a similar β-lactone, the lack of a methyltransferase (cmoA) significantly affects the characteristics of the β-lactone scaffold, as the presence of methyl groups plays a crucial protective role during the initial synthesis of the β-lactone head. This methylation may not be present in the final product, which could explain the large differences in inhibition rates (51).

Chitin is a simple polysaccharide that provides an important component of the carbohydrate backbone of fungal cell walls (52). Chitin is broken down by chitinase,

which breaks the β (1,4)-glycosidic bonds between *N*-acetyl-D-glucosamine units, resulting in the production of monomers and oligosaccharides (53). Bacterial strains with chitinase genes exhibited strong inhibition rates against *P. destructans*. Notably, in *Pseudomonas*, only one strain produced chitinase, and it had a much higher inhibition rate than the other three strains (Table S4). Of note, we also found the highest inhibition rate in the *Acinetobacter*_4 (with only two BGCs), which also contains a linear azol(in)e-containing peptide (LAP) (Table S2). LAPs are a rapidly developing, but under-studied, class of peptides that are synthesized in the ribosome post-translationally modified. Most LAPs are capable of inhibiting DNA helicase or ribosomes, exhibiting cell lysis and antimicrobial properties (54). It is possible that the LAP in this strain could be a key gene cluster for inhibiting the growth of *P. destructans*.

We also identified several known gene clusters. In the *Bacillus*_2 strain, we found a gene cluster homologous to *Bacillus cereus* zwittermicin A biosynthetic genes (Table S5). Zwittermicin A is a linear aminopolyol antibiotic with diverse biological activities, inhibition of fungal and bacterial growth (55–57). Additionally, in the *Pantoea*_1 strain, we predicted a gene cluster homologous to carotenoid (Table S4), which can also exhibit antifungal activity (58).

One limitation of this study is that all inhibition assays were conducted using a single agar type (SDA) and a single incubation temperature (13°C). It is known that the growth and inhibition of *P. destructans* can be influenced by media composition, agar salt content, and pH (59). Therefore, we plan to conduct further studies under different media compositions, pH levels, and other culture conditions to validate the effectiveness of these inhibitory effects.

## Conclusion

In all, our study shows that that several bacterial strains isolated from bat skin and the surrounding soil were able to inhibit the *in vitro* growth of *P. destructans*, by secreting a variety of small molecular metabolites and VOCs. Using whole-genome sequencing, we analyzed the BGCs and their antagonistic influence. We then further characterized these BGCs to identify several key gene clusters that may inhibit the growth of *P. destructans*. To explore the feasibility of using these probiotics and their metabolites to control *P. destructans*, future research should first focus on laboratory-based evaluations of their stability, dosage, biosafety, and ecological impacts on cave ecosystems. Only after successful laboratory validation should field trials be considered to assess their potential for reducing *P. destructans* environmental reservoirs.

## MATERIALS AND METHODS

### *Pseudogymnoascus destructans*

*Pseudogymnoascus destructans* JHCN111a isolates came from the Jilin Provincial Key Laboratory of Animal Resource Conservation and Utilization, Northeast Normal University, China. According to the recent classification proposed by Fischer et al., this isolate belongs to the *Pd*-2 clade (60). Conidia suspension was prepared by adding 3-week-old cultured *P. destructans* on Sabouraud dextrose agar (SDA) to 8 mL of 1 × PBST$_{20}$ buffer. Using a sterile inoculation loop, the conidia were gently scraped from the culture surface and then filtered through sterile cotton. A fresh spore suspension was prepared before each experiment.

### Collection of bat skin microbiota and habitat soil samples

From December 2022 to April 2023, bat skin swabs and soil samples were collected from fo bat roosting caves in Jilin Province (Di Cave, Gezi Cave and New Cave), Liaoning Province (Temple Cave), and Beijing (Bat Cave) (61). After the capture of bats, sterile cotton swabs were moistened with sterile water and gently rolled over the surface of the bats' wing membranes to collect skin microbiota samples. The swab samples were

then placed in a 30% solution of sterile glycerol and stored in a freezer at −80°C. A total of 50 skin samples were collected from four species: *Rhinolophus ferrumequinum*, *Murina leucogaster*, *Myotis petax,* and *M. ricketti*. In addition, 10 soil samples were collected from each cave. Approximately 20 g of cave sediment was collected from the surface layer (5–10 cm) using sterilized stainless steel scoops. Samples were taken from various locations within caves, typically beneath or near bat roosting sites, at a distance of approximately 5–15 m from the cave entrance. Areas with frequent human activity were intentionally avoided, and sampling was restricted to relatively undisturbed zones. The collected sediment primarily consisted of moist clay, containing occasional small amounts of organic matter such as fallen leaves, bat guano, and insect remains, together with a small amount of rock. All samples were placed into sterile 100 mL reagent bottles to prevent external contamination.

## Isolation of bacterial strains

Bacterial strains were isolated following the methods described by Li et al. (11). Skin and soil samples were separately subjected to serial 10-fold dilutions in sterile water. For skin samples, dilutions were prepared to $10^{-1}$ and $10^{-2}$, while soil samples were diluted to $10^{-5}$ and $10^{-6}$. Three dilution levels from each sample were plated to ensure the recovery of bacterial communities with varying abundance. Aliquots of the diluted samples were subsequently inoculated onto R2A agar plates (per liter: 0.5 g yeast extract, 0.5 g proteose peptone, 0.5 g casamino acids, 0.5 g glucose, 0.5 g soluble starch, 0.3 g sodium pyruvate, 0.3 g $K_2HPO_4$, 0.05 g $MgSO_4 \cdot 7H_2O$, and 15 g agar) (62), which is commonly used for isolating slow-growing and oligotrophic bacteria from low-nutrient environments. Plates were incubated at 13°C, the average temperature determined from *in situ* measurements of the cave walls during winter using an infrared thermometer, in order to simulate the natural cave environment and support the recovery of psychrotolerant microorganisms. Colony growth was monitored beginning at 36 h and continued until 72 h to capture colonies in the exponential growth phase when bacterial activity is optimal. Based on morphology and color, single colony was selected and transferred to fresh R2A agar plates. This process was repeated twice until pure strains were obtained.

## Identification of bacterial strains

We identified the isolated bacterial strains using PCR amplification and DNA sequencing. Bacterial genomic DNA was extracted using the Tiangen Bacterial Genomic DNA Extraction Kit (Tiangen Biotech, Beijing, China) following the manufacturer's instructions. The universal bacterial 16S rRNA gene primers 27F and 1492R were used for PCR amplification (63). The PCR products were sequenced, and the resulting sequences were assembled using DNAStar 7.1 software. The assembled sequences were then compared to the 16S rRNA bacterial database from NCBI using BLAST to identify the most similar sequences.

## Agar plate challenge assays

In the experiment, a single-compartment culture dish (200 mm × 30 mm) was used as a shared space to screen bacterial strains for anti-*P. destructans* activity through contact-independent interactions. To do so, 100 µL of *P. destructans* conidia suspension ($2 \times 10^6$ conidia/mL) was evenly spread onto SDA agar plates (90 mm × 18 mm). At the same time, 100 µL of bacterial suspension (cultured at 13°C and 200 r/min for 24–48 h) was inoculated onto Luria–Bertani (LB) agar plates (90 mm×18 mm). LB medium, composed mainly of tryptone (10 g/L), yeast extract (5 g/L), sodium chloride (10 g/L), and agar (15 g/L), is a nutrient-rich medium commonly used for cultivating fast-growing bacteria with low nutritional requirements (64). We chose LB broth to culture the antagonistic bacterial strains because it allows for rapid accumulation of sufficient biomass under stable growth conditions, which is beneficial for the subsequent evaluation of antifungal activity. Next, the two plates were co-incubated within the shared space at 13°C for 14 days to evaluate the fungistatic effects on *P. destructans*.

## Evaluation of the inhibitory effects of volatile organic compounds produced by antagonistic strains

In this subsequent experiment, the previously screened antagonistic bacterial strains and *P. destructans* were cultured separately, without shared-space co-culture. The antagonistic strains were inoculated in LB broth and cultured at 13℃ and 200 r/min shaking for 24–48 h to prepare bacterial suspensions. Next, 100 µL of the antagonistic strain suspension was inoculated onto LB agar plates, while uninoculated LB agar plates served as controls. At the same time, 100 µL of *P. destructans* conidia suspension ($2 \times 10^6$ conidia/mL) was evenly spread onto SDA agar plates, with uninoculated SDA plates as controls. All plates were incubated at 13℃ for 14 days. The purpose of this experiment was to prepare for GC-MS analysis aimed at identifying and confirming the inhibitory effects of specific volatile organic compounds.

Volatile organic compounds (VOCs) produced by anti-*P. destructans* bacterial strains via contact-independent mechanisms were detected using headspace solid-phase microextraction (HS-SPME). Culture plates designated for analysis were placed in headspace vials and incubated at 40℃ for 30 min to allow for VOC accumulation. A 50/30 µm DVB/CAR/pdMS fiber was pre-conditioned in the injection port of the gas chromatography-mass spectrometry (GC-MS) (Agilent 5975, United States) system at 270℃ for 30 min prior to use. The conditioned SPME fiber was inserted into the headspace vial, gently exposed, and suspended in the upper headspace above the sample for 20 min to allow adsorption of volatile compounds. Following adsorption, the fiber was retracted into the needle and introduced into the GC-MS system for analysis. Separation of VOCs was performed using a chromatographic column (60 m × 0.32 mm × 0.25 µm, Agilent 19,091J-416, United States). The column temperature program was as follows: an initial temperature of 50℃ held for 3 min, followed by an increase at a rate of 10℃/min to 280℃, where it was maintained for 20 min, resulting in a total runtime of 46 min. Both the inlet and thermal auxiliary (THERMAL AUX 2) temperatures were set to 260℃. Identification of volatile compounds was achieved by comparing spectral data from the National Institute of Standards and Technology (NIST) database (11). Volatile compounds from antagonistic strains isolated from soil and skin were normalized and subsequently visualized using principal component analysis (PCA). Differences in volatile organic compounds between the two groups were then assessed using permutational multivariate analysis of variance (PERMANOVA) based on a Bray–Curtis distance matrix with the *adonis* function in the R package *vegan*.

To verify the inhibitory effects of VOCs on *P. destructans* growth, 100 µL of *P. destructans* conidia suspension ($2 \times 10^6$ conidia/mL) was evenly spread onto the SDA agar surface at the bottom of a Petri dish (90 mm × 18 mm). Sterile antibiotic susceptibility discs loaded with either 10 µL or 100 µL of one of the following compounds (2-Undecanone, 2-Tridecanone, Benzaldehyde, 1-Undecene, 2-Nonanone, Thujone, 2,5-Dimethylpyrazine, or α-Pinene) were affixed to the inner surface of the Petri dish lid (i.e., opposite the SDA agar surface). *P. destructans* conidia inoculated onto SDA agar plates without VOC exposure served as the negative control. All plates were incubated at 13℃ for 14 days.

## Effects of volatile organic compounds on *P. destructans* mycelial morphology

After 14 days of VOC exposure, *P. destructans* mycelium was collected from the inhibition zone on the SDA agar plates. Mycelium not exposed to VOCs served as the negative control. Samples were fixed at room temperature for 15 min in 2.5% glutaraldehyde, followed by three 10 min washes with 0.01 M PBST buffer. Dehydration was performed sequentially in 30%, 50%, 70%, 80%, and 90% ethanol for 15 min each, followed by three 15 min incubations in absolute ethanol. The samples were then air-dried at room temperature for 18 h and sputter-coated with gold. Morphological observations were conducted using a scanning electron microscope (SEM, FEI XL-30 ESEM-FEG, USA).

## Cell-free supernatant challenge assays

A cell-free supernatant inhibition assay was performed in a 96-well microplate to screen for *P. destructans*-antagonistic bacterial strains that produce water-soluble secondary metabolites. Each bacterial isolate was inoculated into 10 mL of sterile LB broth and incubated with at 13°C with shaking at 200 r/min for 24–48 h. Cultures were centrifuged at 8,000 r/min for 10 min, and the resulting supernatant was collected. *P. destructans* conidia were suspended in 1% tryptone broth to prepare a conidial suspension. In the experimental wells, 50 µL of *P. destructans* conidial suspension ($2 \times 10^6$ conidia/mL) and 50 µL of bacterial cell-free supernatant were added. Positive control wells contained 50 µL of *P. destructans* conidial suspension and 50 µL of 1% tryptone broth. Negative control wells contained 50 µL of heat-inactivated conidia (incubated at 60°C for 45–60 min) and 50 µL of 1% tryptone broth. Each treatment and control group was replicated 12 times. The 96-well plate was incubated at 13°C, and absorbance was measured at 492 nm using a spectrophotometer from day 0 to day 7. The inhibition rate (%) was calculated using the following formula: Inhibition rate = $1 - [(\text{Experimental group}_{OD} - \text{Negative control group}_{OD})/(\text{Positive control group}_{OD} - \text{Negative control group}_{OD})] \times 100\%$. The highest inhibition rate observed during the 7-day period was recorded as the inhibition score for the each bacterial strain.

## Determining activity of protein and non-protein fractions in fermentation broth of antagonistic strains

A total of 10 mL of bacterial fermentation broth (cultured at 13°C with shaking at 200 r/min for 24–48 h) was mixed with 30 mL of methanol and centrifuged at 9,500 r/min for 15 min at room temperature. The resulting supernatant (representing the non-protein fraction) and the protein precipitate were collected. The supernatant was subjected to rotary evaporation at 65°C to remove methanol and then reconstituted with sterile water to the original volume. The protein precipitate was similarly resuspended in sterile water to the original volume. The biological activity of both fractions was subsequently assessed using the cell-free supernatant inhibition assay described above.

## Molecular weight range of active compounds in the fermentation broth of antagonistic strains

A total of 50 mL of the bacterial fermentation broth was transferred into dialysis bags with molecular weight cutoffs of 1,000 Da and 3,000 Da and immersed in 150 mL of deionized water for dialysis at 4°C. The deionized water was replaced every 12 h, for a total of six exchanges. Following dialysis, the contents inside and outside the dialysis bags were concentrated to their original volumes. The biological activities of both fractions were subsequently evaluated using the cell-free supernatant challenge assay.

## Polarity study of fungistatic substances in the fermentation broth of antagonistic strains

A total of 100 mL of the bacterial fermentation broth was centrifuged at 10,000 r/min for 10 min, and the resulting supernatant was collected. The supernatant was sequentially extracted 6–8 times using 75 mL of petroleum ether, ethyl acetate, dichloromethane, and 1-butanol. After extraction, the organic and aqueous phases were separated and concentrated to approximately 5 mL using rotary evaporation. Each extract was then reconstituted with sterile water to the original volume. The biological activities of all fractions were subsequently evaluated using the cell-free supernatant challenge assay.

## Whole-genome sequencing and analysis of antagonistic strains

Strains exhibiting an inhibition rate above 50% and those producing VOCs (*n* = 29) were selected for DNA extraction using the STE method (65). High-throughput sequencing and library construction were performed on the Illumina Novaseq platform. For

each DNA fragment library, 1,500 bp insert fragments were sequenced from both ends. Low-quality reads—those with more than 40% of bases having a quality score ≤20, over 10% ambiguous bases ($N$), or adapter overlaps exceeding 15 bp with fewer than three mismatches—were were removed. Genome assemblies were generated using multiple assemblers. For SOAP denovo, different K-mer sizes (95, 107, 119) were tested, and for assembly with the fewest scaffolds, following parameter adjustments (-d -u -R -F), were selected as the preliminary result. SPAdes was used with K-mer values of 99 and 127, again selecting the assembly with the fewest scaffolds. Abyss was applied using K-mer 64 as a parameter for assembly. Final assemblies from all tools integrated using CISA, prioritizing the version with the lowest scaffold count. Gaps were filled using GapCloser. To reduce potential lane contamination, reads with sequencing depth with less than 35% of the average were filtered out. In addition, assembled fragments shorter than 500 bp were removed. Coding sequences were predicted using GeneMarkS (version 4.17) (66–71). Biosynthetic gene clusters associated with secondary metabolite production were identified using antiSMASH 5.0 (18). Secondary metabolites from antagonistic strains isolated from soil and skin were normalized and subsequently visualized using PCA. Differences in secondary metabolite profiles between the two groups were then evaluated via PERMANOVA based on a Bray–Curtis distance matrix using the *adonis* function in the R package *vegan*. Genes related to VOC biosynthesis were annotated using the Kyoto Encyclopedia of Genes and Genomes (KEGG) database (72).

## ACKNOWLEDGMENTS

This work was supported by the Jilin Provincial Natural Science Foundation (grant no. YDZJ202401501ZYTS) and Nation Natural Science Foundation of China (grant nos. 32300425, 32171525, 31961123001, and 32171481) and Innovation and Entrepreneurship Training Program for College Students in Jilin Province (grant nos. 202310193005 and S202410193089).

S.S. conducted data analysis and wrote the manuscript. S.S., M.S., Z.H., Z.W., Y.L., and M.S. collected the samples. Z.L., K.S., and J.F. reviewed and revised the manuscript.

## AUTHOR AFFILIATIONS

[1]College of Life Science, Jilin Agricultural University, Changchun, China
[2]Jilin Provincial Key Laboratory of Animal Resource Conservation and Utilization, Northeast Normal University, Changchun, China
[3]Jilin Provincial International Cooperation Key Laboratory for Biological Control of Agricultural Pests, Changchun, China

## AUTHOR ORCIDs

Shaopeng Sun http://orcid.org/0009-0005-4016-7547
Zihao Huang https://orcid.org/0009-0005-3407-992X
Keping Sun https://orcid.org/0000-0002-4227-9818
Zhongle Li http://orcid.org/0000-0002-2137-9967
Jiang Feng http://orcid.org/0000-0002-7503-1069

## FUNDING

| Funder | Grant(s) | Author(s) |
| --- | --- | --- |
| Jilin Provincial Natural Science Foundation | YDZJ202401501ZYTS | Zhongle Li |
| Innovation and Entrepreneurship Training Program for College Students in Jilin Province | 202310193005 | Zhongle Li |
| National Natural Science Foundation of China | 31961123001 | Keping Sun |
| National Natural Science Foundation of China | 32171481 | Keping Sun |
| National Natural Science Foundation of China | 32171525 | Keping Sun |

| Funder | Grant(s) | Author(s) |
|---|---|---|
| National Natural Science Foundation of China | 32300425 | Zhongle Li |
| Innovation and Entrepreneurship Training Program for College Students in Jilin Province | S202410193089 | Zhongle Li |

## AUTHOR CONTRIBUTIONS

Shaopeng Sun, Formal analysis, Investigation, Methodology, Resources, Software, Validation, Visualization, Writing – original draft | Mingqi Shan, Data curation | Zihao Huang, Data curation | Yihan Lv, Data curation | Zizhen Wei, Data curation | Mingqi Shen, Data curation | Keping Sun, Funding acquisition, Project administration, Supervision | Zhongle Li, Data curation, Funding acquisition, Investigation, Project administration, Supervision | Jiang Feng, Funding acquisition, Project administration, Supervision

## DATA AVAILABILITY

All raw sequence data are deposited, in the NCBI sequence read archive with accession no. PRJNA1229145.

## ADDITIONAL FILES

The following material is available online.

### Supplemental Material

**Figure S1 (Spectrum01241-25-s0001.tif).** PCA analysis of biosynthetic gene clusters of antagonistic strains against *P. destructans* from bat skin and soil samples.
**Figure S2 (Spectrum01241-25-s0002.tif).** Comparative analysis of biosynthetic gene clusters in the *Bacillus*.
**Tables S1 to S5 (Spectrum01241-25-s0003.docx).** Support information for screening microbial inhibitors of *Pseudogymnoascus destructans* in northern China.

### Open Peer Review

**PEER REVIEW HISTORY (review-history.pdf).** An accounting of the reviewer comments and feedback.

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
