## [Reviewer comments · Microbiology Spectrum]

Microbiology Spectrum

Screening microbial inhibitors of *Pseudogymnoascus destructans* in northern China

Shaopeng Sun, Mingqi Shan, Zihao Huang, Yihan Lv, Zizhen Wei, Mingqi Shen, Keping Sun, Zhongle Li, and Jiang Feng

Corresponding Author(s): Zhongle Li, Jilin Agricultural University 吉林农业大学

Review Timeline:

Submission Date:	April 20, 2025
Editorial Decision:	June 14, 2025
Revision Received:	July 16, 2025
Editorial Decision:	August 17, 2025
Revision Received:	September 8, 2025
Accepted:	September 30, 2025

Editor: Renato Kovacs

Reviewer(s): The reviewers have opted to remain anonymous.

Transaction Report:

DOI: <https://doi.org/10.1128/spectrum.01241-25>

Re: Spectrum01241-25 (Screening microbial inhibitors of *Pseudogymnoascus destructans* in northern China)

Dear Dr. Zhongle Li:

Thank you for the privilege of reviewing your work. Below you will find my comments, instructions from the Spectrum editorial office, and the reviewer comments.

Revision Guidelines

Sincerely,
Renato Kovacs
Editor
Microbiology Spectrum

Reviewer #1 (Comments for the Author):

This work is important, and I believe the overall study design was solid. I would suggest more stringent thresholds for sequencing results, as an N50 of ~7.7kb is well below the accepted standard for bacterial genome assemblies and possibly indicates that the data is not reliable for detailed functional or comparative analyses. If the average is much higher, I suggest including it or a histogram of the contig lengths which answers concerns about the quality of the WGS data.

Reviewer #2 (Comments for the Author):

This manuscript is about bacteria on cave walls and bat skin in China. This is interesting as the site is within the native range of *Pseudogymnoascus destructans*, and the results can make an interesting comparison to similar work done in North America. Throughout my comments I refer to *Pseudogymnoascus destructans* as Pd. There are some grammar mistakes throughout the manuscript. The methods are not provided in enough detail. Abbreviations are not always defined before they are used (media types).

Please incorporate this paper in your study and determine which species of Pd you used: Fischer et al. 2025. Two distinct host-specialized fungal species cause white-nose disease in bats

Line 34-35: "Bat skin and environmental microbes play an important role in reducing *Pseudogymnoascus destructans* infection." What evidence demonstrates this in vivo? Do you mean WNS instead of Pd infection? Does 'Pd infection' in this instance mean that Pd is present in the environment or on the skin? If so, that is not infection.

Lines 38-39: "This study lays the foundation for further exploration and elucidation of the mechanisms by which bacteria from bat skin and roosting environments suppress *P. destructans* growth" Need to add 'in vitro' at the end of the sentence.

Line 51: "Moreover, because these fungal pathogens can persist outside the host for extended periods (5)" Study 5 (Hoyt et al) was done in petri plates in the lab. It does not demonstrate that Pd persists in the environment in the absence of bats.

Line 52: "a potential mitigation approach can be to minimize their environmental loads (6), for example, via the application of environmental probiotics." Study 6 (Hoyt et al) did not demonstrate that probiotics decreased Pd environmental loads.

Lines 54: "White-nose syndrome (WNS) was first identified in 2006 and has caused significant mortality in bat populations (2) (7)" These are outdated references when discussing WNS mortality in North America. Please update.

Lines 59-61: "several microbial inhibitors of *P. destructans* have been identified from the skin of bats in China (e.g., *Pseudomonas*, *Rhodococcus*, and *Acinetobacter*) (10). This could result from the secretion of phenazine-1-carboxylic acid and various volatile organic compounds (VOCs) that suppress *P. destructans* growth (11)". Please clarify that these studies were done in vitro.

Methods:

Lines 3345-346: "Approximately 20 g of soil was collected from the surface layer (5-10 cm) and placed in sterile 100 mL reagent bottles."

Where in caves were samples collected (near to bats? Near to the cave entrance? On well-worn paths of human foot traffic)? Define what soil is in this instance as caves are not considered to have soil due to the lack of organic matter. Was it clay? Was it wet or dry? Was it rocky? Did it have bits of leaves, guano, or other organic matter? What was used to collect the samples?

Lines 348-349: The skin and soil samples were diluted with sterile water and then inoculated onto R2A agar plates, which were incubated at 13C for 36-72 h."

How much were samples diluted? Were multiple dilutions from one swab plated? Explain why R2A agar was chosen and link to the recipe. Why were bacterial plates incubated at 13C: how was that temperature chosen (also line 368)? Why were plates incubated for 36-72 hours: how was the timing chosen?

Line 365: why was SDA chosen? Link to the recipe for this agar type. Did you try different types of agar to see if your results changed? Whether Pd is inhibited or not depends on the media type it is grown on, agar salt content, and agar pH. Therefore, lab results may not be applicable in vivo. The outcome of interactions among microbes depends on environmental conditions. How much do these media types and conditions resemble the natural substrates Pd grows on? Were inhibition assays done at different temperatures? Why or why not?

Line 367: define what LB agar is and why it was chosen

Lines 362-369 and 372-378 appear to describe the same experiment- please clarify how they differ

Line 227: "These VOCs are low in toxicity and fast acting, and can be effective in alleviating WNS through aerosol diffusion (39)." This statement is incorrect and not supported by the literature. Black Diamond Tunnel in northwestern Georgia received an anti-fungal treatment from 2011 through 2022 (Gabriel et al., 2022). However, bats in this hibernaculum still had high Pd loads during this study (Ferrari 2022). Therefore the VOC treatments did not reduce bats' exposure to this pathogen. Ferrari, M. Investigating biometrics of *Perimyotis subflavus* in traditional and nontraditional hibernacula in the Southeastern United States as it relates to susceptibility to white-nose syndrome. M.S. Thesis, Kennesaw State University. (2022).

Lines 322-325: "To validate the practical application of these probiotics and their active substances in natural environments, next

steps would be to conduct field experiments on the active compounds to assess their stability, biosafety, and long-term potential for controlling *P. destructans* infections. This holds significant implications for the prevention and control of WNS."

Is this suggesting that bats should be sprayed with bacteria/VOCs or the cave itself? Are you trying to treat bats directly or reduce the environmental reservoir of Pd? If you are trying to treat bats directly, there is no evidence that bacteria/VOCs will help once the fungus is deep in tissues - would the bacteria/VOCs be able to reach the fungus or have an effect? Do bats in Eurasia need to be treated? On what continent should these bacteria/VOCs be deployed? What effect would spraying bacteria/VOCs in caves have on other vertebrates, invertebrates, and microbes in caves? What is the minimum dosage needed? It is premature to call for field experiments when no safety testing has been done in the lab.

Additionally, Pd environmental reservoirs are disappearing in eastern North America with no human intervention

<https://esajournals.onlinelibrary.wiley.com/doi/full/10.1002/ecs2.70149>

Screening microbial inhibitors of *Pseudogymnoascus destructans* in northern China
Spectrum01241-25
06/13/2025

Major Finding:

- While I do believe that the overall study design is reliable for this kind of assessment, I strongly advise that this author re-filter the whole-genome sequencing data used to identify the potential antifungal gene clusters. The author doesn't include a figure/histogram to show the distribution of N50 contig lengths, which possibly indicates that a large percentage fall below the accepted standard for bacterial genome assemblies, raising questions of the subsequent biological interpretation of the results.
- If the author provides a histogram that shows that the average was between 100,000 and 500,000 bp, then it's possible that re-filtering isn't necessary, but the authors didn't report an average, raising concerns about the transparency of the quality of data presented in this paper.

Line 157: An N50 of ~7.7Kb is well below the accepted standard for bacterial WGS data and possibly indicates that the data is not reliable for detailed functional or comparative analyses, undermining the subsequent biological interpretation of these results. To address this, the author needs to either include an average or histogram figure that clearly indicates the quality of the WGS data. If a large percentage of the WGS contigs are that low, those need to be filtered out and the analyses need to be redone.

Line 343: Inconsistent naming. Include entire scientific name for all or none.

Line 343: While it's not the point of this study to correlate Pd load with the microbiome of each bat, some additional metadata – isolates from each bat species, corresponding Pd load when sampled (if available) – would be useful to the larger point of this study: the mitigation of Pd and its impact on bat populations. Skin microbiomes are host-specific and knowing which isolates appeared on which bat species would be valuable for the application of this work in future.

Dear Dr. Renato Kovacs ,

We are very appreciative of two reviewers' constructive comments on our manuscript entitled "Screening microbial inhibitors of *Pseudogymnoascus destructans* in northern China" (manuscript number: Spectrum01241-25). Those comments and suggestions are very professional and useful, providing great help for improving our manuscript. We have revised the manuscript carefully and seriously according to the comments and suggestions point by point. The revisions in the new manuscript have been marked in blue with the revised contents. The following contents indicated the detailed changes that we have made in the revised manuscript.

After revision, we think the manuscript has been improved greatly, and we wish to resubmit it to Microbiology Spectrum again. Your consideration and comments will be highly appreciated.

We are looking forward to receiving your kind reply. Many thanks again for all of your work.

With the best regards,

Yours sincerely,

Zhongle Li

College of Life Science, Jilin Agricultural University, Changchun 130118, China

Reviewer: 1

Line 157: An N50 of ~7.7Kb is well below the accepted standard for bacterial WGS data and possibly indicates that the data is not reliable for detailed functional or comparative analyses, undermining the subsequent biological interpretation of these results. To address this, the author needs to either include an average or histogram figure that clearly indicates the quality of the WGS data. If a large percentage of the WGS contigs are that low, those need to be filtered out and the analyses need to be redone.

Response: Thank you very much for your valuable comments. We have thoroughly reviewed all 29 bacterial genome sequencing datasets and confirmed that only one strain had an N50 value of approximately 7.7 kb. This sample has been excluded due to its low quality, and the relevant analyses and descriptions have been revised accordingly in the main text. For the remaining 28 strains, we have added a histogram of the N50 values in the revised manuscript to more clearly illustrate the overall assembly quality. The average N50 value of these genomes is 174,527 bp, which meets the general quality standards for bacterial whole-genome assemblies.

Line 343: Inconsistent naming. Include entire scientific name for all or none.

Response: Thank you very much for your correction. We have revised line 315 in the manuscript, changing “*M. petax*” to the full scientific name “*Myotis petax*” to ensure consistency and standardization in nomenclature.

Line 343: While it’s not the point of this study to correlate Pd load with the microbiome of each bat, some additional metadata – isolates from each bat species, corresponding Pd load when sampled (if available) – would be useful to the larger

point of this study: the mitigation of Pd and its impact on bat populations. Skin microbiomes are host-specific and knowing which isolates appeared on which bat species would be valuable for the application of this work in future.

Response: Thank you very much for your suggestion. We fully agree on the importance of host specificity in skin microbiota and the associated *P. destructans* loads information. Unfortunately, due to missing *P. destructans* loads data for some bat species in our current sample, we are unable to provide complete information at this time. However, we have included detailed information on the isolates from each species in manuscript (Table 1 and 2). We will continue to improve and expand this dataset in our future work.

Reviewer: 2

Please incorporate this paper in your study and determine which species of Pd you used: Fischer et al. 2025. Two distinct host-specialized fungal species cause white-nose disease in bats.

Response: Thank you very much for your suggestion. We have incorporated the study by Fischer et al. (2025) into the manuscript and clarified the taxonomic affiliation of the Pd isolate used in our study. Specifically, we have added the following sentence to lines 301–302: “According to the recent classification proposed by Fischer et al., this isolate belongs to the *Pd-2* clade (61).” In addition, we have included the full citation in the reference list at lines 652–654: Fischer et al. 2025. Two distinct host-specialized fungal species cause white-nose disease in bats. These revisions ensure that the taxonomic identity of the fungal isolate used in our study is clearly defined and appropriately referenced.

Line 34-35: "Bat skin and environmental microbes play an important role in reducing *Pseudogymnoascus destructans* infection." What evidence demonstrates this *in vivo*? Do you mean WNS instead of Pd infection? Does 'Pd infection' in this instance mean that Pd is present in the environment or on the skin? If so, that is not infection.

Response: Thank you very much for your valuable comments. We agree that the use of the term “*Pseudogymnoascus destructans* infection” in this context may be ambiguous. We have revised the sentence to clarify that it refers to the presence of *Pseudogymnoascus destructans* on bat skin or in the environment, rather than clinical infection. In this context, the original use of “Pd infection” was intended to broadly indicate the presence of Pd, but we recognize that a more accurate term would be white-nose syndrome (WNS). Therefore, we have revised the sentence on lines 34–36 as follows: “Bat skin and environmental microbiota may influence the colonization and persistence of *Pseudogymnoascus destructans*, thereby potentially affecting the occurrence of white-nose syndrome.”

We also acknowledge that *in vivo* evidence for this mechanism remains limited, and the statement is primarily supported by studies *in vitro*. We appreciate your help in improving the clarity and scientific accuracy of our manuscript. Our results show that certain bacteria isolated from bat skin or cave environments exhibit anti-Pd activity *in vitro*, supporting this hypothesis, though further *in vivo* validation is needed.

Lines 38-39: "This study lays the foundation for further exploration and elucidation of the mechanisms by which bacteria from bat skin and roosting environments suppress

P. destructans growth" Need to add 'in vitro' at the end of the sentence.

Response: Thank you to the reviewer for the thorough review and valuable suggestions. We have added the term “*in vitro*” at the end of the sentence in line 40 to improve accuracy:

“This study lays the foundation for further exploration and elucidation of the mechanisms by which bacteria from bat skin and roosting environments suppress *P. destructans* growth *in vitro*.”

Line 51: "Moreover, because these fungal pathogens can persist outside the host for extended periods (5)" Study 5 (Hoyt et al) was done in petri plates in the lab. It does not demonstrate that Pd persists in the environment in the absence of bats.

Response: Thank you very much for your valuable comments. Indeed, the study by Hoyt et al. was conducted primarily under laboratory conditions using petri dishes. While it provides useful information about the growth of *Pseudogymnoascus destructans* (Pd) under these specific conditions, it does not offer evidence regarding the persistence of Pd in natural environments without the presence of bats. Therefore, citing this study does not adequately support the point in question.

To address this issue, we have replaced the reference with the study by Lorch et al. (2013), titled “Distribution and Environmental Persistence of the Causative Agent of White-Nose Syndrome, *Geomyces destructans*, in Bat Hibernacula of the Eastern United States.” This study specifically investigates the environmental persistence of *P. destructans* in bat hibernacula and provides strong field-based evidence demonstrating *P. destructans* persistence under natural conditions, which aligns more closely with the environmental context discussed in our manuscript.

Line 52: "a potential mitigation approach can be to minimize their environmental loads (6), for example, via the application of environmental probiotics." Study 6 (Hoyt et al) did not demonstrate that probiotics decreased Pd environmental loads.

Response: Thank you for your valuable feedback. Upon further review, we confirmed that Study 6 (Hoyt et al.) does not provide direct evidence that probiotics reduce the Pd environmental loads. To better align with the context of our study, this reference has been replaced with Amanpreet et al. (2018), “*Trichoderma polysporum* selectively inhibits white-nose syndrome fungal pathogen *Pseudogymnoascus destructans* amidst soil microbes.” This study investigates how *Trichoderma polysporum* selectively suppresses the white-nose syndrome fungal pathogen Pd within soil microbial communities, thereby reducing Pd loads in the soil. This work is more directly related

to environmental mitigation of Pd, highlighting the potential of soil microbiota in managing fungal pathogen loads.

Lines 54: "White-nose syndrome (WNS) was first identified in 2006 and has caused significant mortality in bat populations (2) (7)" These are outdated references when discussing WNS mortality in North America. Please update.

Response: Thank you very much for your valuable suggestion. We acknowledge that the references regarding the mortality rate of white-nose syndrome (WNS) in North America are somewhat outdated. We have searched for and incorporated the latest relevant studies and statistical data to ensure the citations are timely and accurate. The text in lines 54–55 has been revised to: "White-nose syndrome (WNS) was first identified in 2006 and has caused significant mortality in bat populations." Additionally, we have cited a more comprehensive and recent study: Cheng et al. 2021, The scope and severity of white-nose syndrome on hibernating bats in North America, which has been added at lines 493–495 to reflect the current understanding of WNS impacts on bats.

Lines 59-61: "several microbial inhibitors of *P. destructans* have been identified from the skin of bats in China (e.g., *Pseudomonas*, *Rhodococcus*, and *Acinetobacter*) (10). This could result from the secretion of phenazine-1-carboxylic acid and various volatile organic compounds (VOCs) that suppress *P. destructans* growth (11)". Please clarify that these studies were done *in vitro*.

Response: We sincerely appreciate your detailed suggestions. We have added clarifications in lines 60–62, stating that the inhibitory effects of *Pseudomonas*, *Rhodococcus*, and *Acinetobacter* on *P. destructans*, as well as the functions of their secreted phenazine-1-carboxylic acid and volatile organic compounds (VOCs), were all studied under *in vitro* conditions. This addition will help readers better understand the research context and results more accurately.

Methods:

Lines 345-346: "Approximately 20 g of soil was collected from the surface layer (5-10 cm) and placed in sterile 100 mL reagent bottles."

Where in caves were samples collected (near to bats? Near to the cave entrance? On well-worn paths of human foot traffic)? Define what soil is in this instance as caves are not considered to have soil due to the lack of organic matter. Was it clay? Was it wet or dry? Was it rocky? Did it have bits of leaves, guano, or other organic matter?

What was used to collect the samples?

Response: Thank you very much for your valuable comments. The “soil” samples in this study were collected from various locations within caves, typically beneath or near bat roosting sites, at a distance of approximately 5–15 m from the cave entrance. This ensured that the sampling areas were within bat activity zones rather than near the entrance. We intentionally avoided paths with frequent human activity and prioritized relatively undisturbed sediment areas for sampling. The collected samples consisted of moist clay. The clay surface occasionally contained small amounts of organic matter such as fallen leaves, bat guano, and insect remains. Sampling was conducted using stainless steel scoops that had been sterilized by autoclaving, along with sterile containers, to ensure that the samples were free from external contamination.

Lines 348-349: The skin and soil samples were diluted with sterile water and then inoculated onto R2A agar plates, which were incubated at 13°C for 36-72 h."

How much were samples diluted? Were multiple dilutions from one swab plated? Explain why R2A agar was chosen and link to the recipe. Why were bacterial plates incubated at 13°C: how was that temperature chosen (also line 368)? Why were plates incubated for 36-72 hours: how was the timing chosen?

Response: Thank you for your detailed question. We primarily referred to the isolation methods described in Zhongle Li et al. (2021), Activity of bacteria isolated from bats against *Pseudogymnoascus destructans* in China. Accordingly, we have added the following clarifications to the Methods section:

Dilution details: Skin and soil samples were separately serially diluted 10-fold in sterile water. Skin samples were diluted to 10^{-1} and 10^{-2} , while soil samples were diluted to 10^{-5} and 10^{-6} . Two dilution levels from each sample were plated to ensure recovery of bacterial communities with varying abundance .

Choice of R2A medium: R2A medium is well-suited for isolating slow-growing and oligotrophic bacteria from low-nutrient environments such as caves. It is widely used in environmental microbiology research. We have added a reference link to the medium formulation.

Recipe link: https://en.wikipedia.org/wiki/R2A_agar

Incubation temperature (13°C): Plates were incubated at 13°C to simulate natural cave conditions (typically 10–14°C), which supports the recovery of microorganisms adapted to low temperatures.

Incubation duration (36–72 hours): The extended incubation period reflects the slower

microbial growth at lower temperatures. Colony formation was monitored starting at 36 h and continued until 72 h, aiming to capture the exponential growth phase when bacterial activity is optimal.

We have revised lines 320–323 as follows: Skin and soil samples were separately subjected to 10-fold serial dilutions in sterile water. Skin samples were diluted to 10^{-1} and 10^{-2} , and soil samples to 10^{-5} and 10^{-6} . Aliquots of the diluted samples were subsequently inoculated onto R2A agar plates.

Line 365: why was SDA chosen? Link to the recipe for this agar type. Did you try different types of agar to see if your results changed? Whether Pd is inhibited or not depends on the media type it is grown on, agar salt content, and agar pH. Therefore, lab results may not be applicable in vivo. The outcome of interactions among microbes depends on environmental conditions. How much do these media types and conditions resemble the natural substrates Pd grows on? Were inhibition assays done at different temperatures? Why or why not?

Response: Thank you very much for raising this important question. Our methods for culturing *Pseudogymnoascus destructans* (Pd) were primarily based on Li et al. (2021), Activity of bacteria isolated from bats against *Pseudogymnoascus destructans* in China, and Adrian et al. (2022), Microbial isolates with Anti-*Pseudogymnoascus destructans* activities from Western Canadian bat wings. We selected Sabouraud Dextrose Agar (SDA) as the base medium, as it is widely used for fungal cultivation—including Pd—due to its good growth performance and suitability for observing antimicrobial effects. The inhibition assays were conducted at 13°C to simulate the typical temperature conditions of cave environments where Pd grows. We have added the medium formulation details.

Recipe link: https://en.wikipedia.org/wiki/Sabouraud_agar

In E. Donaldson et al. (2018), Growth media and incubation temperature alter the Pd transcriptome: implications in identifying virulence factors, a comparison was made between PDA and SDA media. Pd growth was found to be influenced by nitrogen source: SDA contains animal-derived nitrogen, while PDA contains plant-derived nitrogen. SDA more closely resembles the nitrogen sources present in bat hibernacula (e.g., guano, humic substances, insect remains). In that study, Pd cultured on SDA produced more secretions related to host tissue degradation, while morphological changes were observed when grown on PDA. Therefore, SDA provides a culture environment that better mimics host-associated conditions.

We have not yet tested Pd growth on different types of agar, and thus do not know whether inhibitory effects are influenced by factors such as medium type, salt content, or pH. In future work, we plan to consider these variables and evaluate their impact on inhibition outcomes.

Indeed, we acknowledge that in vitro results may not always reflect in vivo conditions, as microbial interactions are highly dependent on environmental factors. While we have attempted to simulate cave-like conditions as closely as possible for Pd growth, it is not feasible to fully replicate the diversity and complexity of natural organic substrates. As a next step, we plan to conduct inhibition experiments across different temperature gradients to assess variations in suppression effects.

Line 367: define what LB agar is and why it was chosen

Response: Thank you very much for your valuable comments. Luria–Bertani (LB) agar is a nutrient-rich medium commonly used for cultivating fast-growing bacteria with low nutritional requirements. It is mainly composed of tryptone, yeast extract, and sodium chloride, which support rapid bacterial growth and the formation of stable colony morphology.

In this study, we primarily referred to the method used in Li et al. (2021), Activity of bacteria isolated from bats against *Pseudogymnoascus destructans* in China, where LB liquid medium was used for bacterial cultivation. We chose LB liquid medium to culture the antagonistic bacterial strains because it allows for the rapid accumulation of sufficient biomass while maintaining good growth conditions, which is beneficial for the subsequent evaluation of antifungal activity.

Recipe link:

<https://www.sigmaaldrich.com/US/en/product/sigma/l3147?srsId=AfmBOooN0fdDtRe6AEW5kHF1BF4BzV-oaInxoOqEUuzJ5h-suGhJz6AJ>

Lines 362-369 and 372-378 appear to describe the same experiment- please clarify how they differ

Response: Thank you very much for your review and comments. We agree that these two experimental descriptions may appear similar on the surface, but they are in fact two independent experiments with different objectives and procedures:

Lines 362–369 describe a shared-space co-culture plate assay designed to screen for anti-*P. destructans* bacterial strains using a non-contact inhibition approach.

Lines 372–378 refer to a subsequent experiment that does not involve shared-space

co-culture. In this case, the previously screened antagonistic bacterial strains and *P. destructans* were cultured separately for subsequent GC-MS analysis, with the primary goal of identifying and verifying the inhibitory effects of specific volatile organic compounds.

Line 227: "These VOCs are low in toxicity and fast acting, and can be effective in alleviating WNS through aerosol diffusion (39)." This statement is incorrect and not supported by the literature. Black Diamond Tunnel in northwestern Georgia received an anti-fungal treatment from 2011 through 2022 (Gabriel et al., 2022). However, bats in this hibernaculum still had high Pd loads during this study (Ferrari 2022). Therefore the VOC treatments did not reduce bats' exposure to this pathogen. Ferrari, M. Investigating biometrics of *Perimyotis subflavus* in traditional and nontraditional hibernacula in the Southeastern United States as it relates to susceptibility to white-nose syndrome. M.S. Thesis, Kennesaw State University. (2022).

Response: Thank you very much for pointing out the inaccuracy in our description of the effects of VOCs. We agree that there is currently no conclusive evidence showing that VOCs can effectively reduce Pd loads in bats or alleviate white-nose syndrome (WNS) under field conditions. Following the reviewer's suggestion, we reviewed and referenced the findings of Gabriel et al. (2022) and Ferrari (2022), and have removed the original statement claiming that VOCs "can be effective in alleviating WNS through aerosol diffusion," as it was not supported by confirmed evidence.

We have revised lines 203-206 as follows: These volatile organic compounds have low toxicity, rapid reaction, potent antifungal activity, and can diffuse via aerosols, indicating potential application value. However, their practical use in natural environments still requires further validation.

Lines 322-325: "To validate the practical application of these probiotics and their active substances in natural environments, next steps would be to conduct field experiments on the active compounds to assess their stability, biosafety, and long-term potential for controlling *P. destructans* infections. This holds significant implications for the prevention and control of WNS."

Is this suggesting that bats should be sprayed with bacteria/VOCs or the cave itself? Are you trying to treat bats directly or reduce the environmental reservoir of Pd? If you are trying to treat bats directly, there is no evidence that bacteria/VOCs will help once the fungus is deep in tissues - would the bacteria/V

OCs be able to reach the fungus or have an effect? Do bats in Eurasia need to be treated? On what continent should these bacteria/VOCs be deployed? What effect would spraying bacteria/VOCs in caves have on other vertebrates, invertebrates, and microbes in caves? What is the minimum dosage needed? It is premature to call for field experiments when no safety testing has been done in the lab.

Additionally, Pd environmental reservoirs are disappearing in eastern North America with no human intervention <https://esajournals.onlinelibrary.wiley.com/doi/full/10.1002/ecs2.70149>

Response: Thank you for your insightful comments. We have revised the relevant sections to clarify that the current study does not propose direct treatment of bats, nor does it intend to pursue immediate field applications. Our objective is to evaluate, following rigorous laboratory validation, whether certain biocontrol strains have the potential to reduce Pd environmental loads in hibernacula.

To address your concerns, we have explicitly clarified the following points:

Target of application: Our envisioned approach involves treating environmental substrates within caves, not the bats themselves.

Stage of research: We agree that field trials are premature without thorough laboratory-based biosafety and ecotoxicological testing. The manuscript text has been revised to emphasize this.

Geographic focus: Our research remains focused on North America. Although there are signs of declining Pd environmental reservoirs in some areas, the fungus has not been eradicated and may resurge.

Eurasian bats: We are not proposing interventions for Eurasian bat populations, where WNS is far less severe.

Ecological impact: We fully agree that any biocontrol strategy must be rigorously assessed for potential effects on cave microbiota and other cave-dwelling organisms (e.g., invertebrates and vertebrates). This point has been clearly stated in the revised manuscript.

Dosage and delivery methods: These parameters remain undefined and will be further explored and optimized in future laboratory studies. No field applications will be considered prior to that.

We have revised lines 291–296 as follows:

“To explore the feasibility of using these probiotics and their metabolites to control *P. destructans*, future research should first focus on laboratory-based evaluations of their

stability, dosage, biosafety, and ecological impacts on cave ecosystems. Only after successful laboratory validation should field trials be considered to assess their potential for reducing Pd environmental reservoirs.”

Re: Spectrum01241-25R1 (Screening microbial inhibitors of *Pseudogymnoascus destructans* in northern China)

Dear Dr. Zhongle Li:

Thank you for the privilege of reviewing your work. Below you will find my comments, instructions from the Spectrum editorial office, and the reviewer comments.

Revision Guidelines

Sincerely,
Renato Kovacs
Editor
Microbiology Spectrum

Reviewer #2 (Comments for the Author):

The authors did not address most of my comments by changing the manuscript, consequently many of my comments are repeated. Information is provided in the 'response to reviewers' that is not incorporated into the manuscript. This information needs to be available as other readers may have the same questions and concerns. Citations to the recipes used for all agar types needs to be provided in the manuscript; Wikipedia is not adequate. The methods are still unclear and lacks key information (see below). The methods contain no information on the statistical analyses that were done (e.g., results presented

on line 105, 148, Fig2). Caveats of the work are not discussed.

Lines 84-85: "This research ... offers novel insights into the treatment of fungal pathogen infections in wildlife." This research did not study fungal pathogen infections in wildlife or demonstrate potential treatments that resolve fungal infections.

Line 114: what does 'significant' mean in this context? Was a statistical test done?

Line 317: "Approximately 20g of soil was collected from the surface layer (5-10 cm) and placed in sterile 100 mL reagent bottles." This requires much more detail to be reproducible. Where in caves were samples collected (near to bats? Near to the cave entrance? On well-worn paths of human foot traffic?)? Define what soil is in this instance as caves are not considered to have soil due to the lack of organic matter. Was it clay? Was it wet or dry? Was it rocky? Did it have bits of leaves, guano, or other organic matter? What was used to collect the samples? All of this information needs to be in the manuscript.

Lines 321-323: "Skin and soil samples were separately subjected to 10-fold serial dilutions in sterile water. Skin samples were diluted to 10^{-1} and 10^{-2} , and soil samples to 10^{-5} and 10^{-6} . Aliquots of the diluted samples were subsequently inoculated onto R2A agar plates. which were incubated at 13{degree sign}C for 36-72 h." How much were samples diluted? Were multiple dilutions from one swab plated? Explain why R2A agar was chosen and link to the recipe. Why were bacterial plates incubated at 13C: how was that temperature chosen (also line 368)? If it is the temperature of the caves, how was that determined? Why were plates incubated for 36-72 hours, that is very short: how was the timing chosen? All of this information needs to be in the manuscript.

Line 341: LB is never defined and it is unclear what this is or why it is used.

Lines 336- 343 and lines 346-352 appear to be describing the same experiment. It is unclear what was done as written.

Paragraph 371- this is a contact-dependent experiment, so it is not necessarily testing VOCs.

Whether Pd is inhibited or not depends on the media type it is grown on, agar salt content, and agar pH. This was demonstrated in Vanderwolf et al. 2021. Skin fungal assemblages of bats vary based on susceptibility to white-nose syndrome. The ISME Journal. The fact that only 1 agar type and 1 temperature was chosen to conduct inhibition assays is a caveat of the study. It is unclear whether in vitro results apply in vivo.

Dear Dr. Renato Kovacs ,

We are very appreciative of reviewer' constructive comments on our manuscript entitled "Screening microbial inhibitors of *Pseudogymnoascus destructans* in northern China" (manuscript number: Spectrum01241-25R1). Those comments and suggestions are very professional and useful, providing great help for improving our manuscript. We have revised the manuscript carefully and seriously according to the comments and suggestions point by point. The revisions in the new manuscript have been marked in blue with the revised contents. The following contents indicated the detailed changes that we have made in the revised manuscript.

After revision, we think the manuscript has been improved greatly, and we wish to resubmit it to Microbiology Spectrum again. Your consideration and comments will be highly appreciated.

We are looking forward to receiving your kind reply. Many thanks again for all of your work.

With the best regards,

Yours sincerely,

Zhongle Li

College of Life Science, Jilin Agricultural University, Changchun 130118, China

Reviewer: 2

The authors did not address most of my comments by changing the manuscript, consequently many of my comments are repeated. Information is provided in the 'response to reviewers' that is not incorporated into the manuscript. This information needs to be available as other readers may have the same questions and concerns. Citations to the recipes used for all agar types needs to be provided in the manuscript; Wikipedia is not adequate. The methods are still unclear and lacks key information (see below). The methods contain no information on the statistical analyses that were done (e.g., results presented on line 105, 148, Fig2). Caveats of the work are not discussed.

Response:

We thank the reviewer for the detailed comments. We have revised the manuscript to incorporate all relevant information previously provided in the “response to reviewers,” so that readers can fully understand the experimental design and rationale.

Specifically:

Agar medium recipes: We have now provided authoritative references and detailed instructions for all agar types used in the study, replacing the Wikipedia links.

Statistical analyses: The comparisons of volatile organic compounds (VOCs) and secondary metabolites among skin and soil bacterial isolates were conducted using principal component analysis (PCA). We have added a description of this method in the Methods section, including the software used. We have added the following sentence at lines 414–419: “Volatile compounds from antagonistic strains isolated from soil and skin were normalized and subsequently visualized using principal component analysis (PCA). Differences in volatile organic compounds between the two groups were then assessed using permutational multivariate analysis of variance (PERMANOVA) based on a Bray–Curtis distance matrix with the *adonis* function in the R package *vegan*.” We have added the following sentence at lines 506–510: “Secondary metabolites from antagonistic strains isolated from soil and skin were normalized and subsequently visualized using PCA. Differences in secondary metabolite profiles between the two groups were then evaluated via PERMANOVA based on a Bray–Curtis distance matrix using the *adonis* function in the R package *vegan*.”

We provided a complete description of the results in lines 111–114: Permutational multivariate analysis of variance (PERMANOVA) based on the Bray–Curtis similarity matrix indicated a significant difference in the relative abundances of

volatile organic compounds (>1%) between groups (Pseudo- $F_{2, 19} = 2.67$, $R^2 = 0.14$, $P = 0.002$; Fig. 2).

We also provided a complete description of the results in lines 156–159: PERMANOVA based on the Bray-Curtis similarity matrix indicated that there was no significant difference in BGC types and abundances between skin and soil samples (Pseudo- $F_{2, 28} = 2.23$, $R^2 = 0.08$, $P = 0.055$; Fig. S1).

Method clarity: All sampling procedures, dilution schemes, incubation conditions, and experimental setups (both contact and non-contact assays) have been clarified in the Methods section.

Study caveats: We have added a Discussion subsection outlining the limitations of this study, including the use of a single agar type and incubation temperature, and the uncertainty of translating *in vitro* results to *in vivo* conditions.

We believe these revisions address the reviewer's concerns and make the Methods and Discussion sections fully comprehensible and reproducible.

Lines 84-85: "This research ... offers novel insights into the treatment of fungal pathogen infections in wildlife." This research did not study fungal pathogen infections in wildlife or demonstrate potential treatments that resolve fungal infections.

Response: Thank you very much for your valuable comments. We agree that our study does not directly demonstrate treatment of fungal infections in wildlife. Our intention was to highlight the potential application of our findings to the management of *Pseudogymnoascus destructans*. To clarify this, we have revised the sentence at Lines 91–93 to: "This research provides scientific evidence supporting the potential development of microbe-based biological control agents against *P. destructans*, offering novel insights into the mitigation of WNS in bats."

This revision more accurately reflects the scope and contribution of our study.

Line 114: what does 'significant' mean in this context? Was a statistical test done?

Response: We thank the reviewer for pointing this out. In this context, "significant" was intended to describe the clear morphological differences observed under scanning electron microscopy, rather than statistical significance. No statistical test was performed on the SEM images. To avoid confusion, we have revised the sentence at Lines 123-124 to: "Scanning electron microscopy revealed visible morphological

changes in *P. destructans* mycelia 14 days after exposure to VOCs.”

This revision more accurately conveys our observation.

Line 317: "Approximately 20g of soil was collected from the surface layer (5-10 cm) and placed in sterile 100 mL reagent bottles." This requires much more detail to be reproducible. Where in caves were samples collected (near to bats? Near to the cave entrance? On well-worn paths of human foot traffic)? Define what soil is in this instance as caves are not considered to have soil due to the lack of organic matter. Was it clay? Was it wet or dry? Was it rocky? Did it have bits of leaves, guano, or other organic matter? What was used to collect the samples? All of this information needs to be in the manuscript.

Response: Thank you very much for your valuable comments. We have revised the description of the sampling procedure at Lines 332-341 to provide sufficient detail for reproducibility. The revised text now reads as follows:

“Approximately 20 g of cave sediment was collected from the surface layer (5–10 cm) using sterilized stainless steel scoops. Samples were taken from various locations within caves, typically beneath or near bat roosting sites, at a distance of approximately 5–15 m from the cave entrance. Areas with frequent human activity were intentionally avoided, and sampling was restricted to relatively undisturbed zones. The collected sediment primarily consisted of moist clay, containing occasional small amounts of organic matter such as fallen leaves, bat guano, and insect remains, together with a small amount of rock. All samples were placed into sterile 100 mL reagent bottles to prevent external contamination.”

Lines 321-323: "Skin and soil samples were separately subjected to 10-fold serial dilutions in sterile water. Skin samples were diluted to 10^{-1} and 10^{-2} , and soil samples to 10^{-5} and 10^{-6} . Aliquots of the diluted samples were subsequently inoculated onto R2A agar plates. which were incubated at 13 {degree sign}C for 36-72 h." How much were samples diluted? Were multiple dilutions from one swab plated? Explain why R2A agar was chosen and link to the recipe. Why were bacterial plates incubated at 13C: how was that temperature chosen (also line 368)? If it is the temperature of the caves, how was that determined? Why were plates incubated for 36-72 hours, that is very short: how was the timing chosen? All of this information needs to be in the manuscript.

Response: Thank you very much for your detailed and constructive comments. In response, we have substantially revised the Methods section (Lines 343–357) to provide more information on sample dilutions, the rationale for choosing R2A agar, incubation temperature, and incubation time. The revised text now reads as follows:

“Bacterial strains were isolated following the methods described by Li et al. Skin and soil samples were separately subjected to serial 10-fold dilutions in sterile water. For skin samples, dilutions were prepared to 10^{-1} and 10^{-2} , while soil samples were diluted to 10^{-5} and 10^{-6} . Three dilution levels from each sample were plated to ensure the recovery of bacterial communities with varying abundance. Aliquots of the diluted samples were subsequently inoculated onto R2A agar plates (per liter: 0.5 g yeast extract, 0.5 g proteose peptone, 0.5 g casamino acids, 0.5 g glucose, 0.5 g soluble starch, 0.3 g sodium pyruvate, 0.3 g K_2HPO_4 , 0.05 g $MgSO_4 \cdot 7H_2O$, and 15 g agar), which is commonly used for isolating slow-growing and oligotrophic bacteria from low-nutrient environments. Plates were incubated at 13 °C, the average temperature determined from *in situ* measurements of the cave walls during winter using an infrared thermometer, in order to simulate the natural cave environment and support the recovery of psychrotolerant microorganisms. Colony growth was monitored beginning at 36 h and continued until 72 h to capture colonies in the exponential growth phase when bacterial activity is optimal.

We have added the following reference at lines 713–714: Reasoner D, Geldreich E. 1985. A new medium for the enumeration and subculture of bacteria from potable water. *Appl Environ Microbiol* 49:1–7.

Line 341: LB is never defined and it is unclear what this is or why it is used.

Response: Thank you very much for your valuable comment. We have revised the Methods section to define LB as Luria–Bertani medium and clarify the reason for its use. LB is a nutrient-rich medium mainly composed of tryptone (10 g/L), yeast extract (5 g/L), and sodium chloride (10 g/L), commonly used for cultivating fast-growing bacteria with low nutritional requirements. In this study, we followed the method described in Li et al. (2021), where LB broth was used for bacterial cultivation. LB broth was chosen because it allows the rapid accumulation of sufficient bacterial biomass under stable growth conditions, which is beneficial for the subsequent evaluation of antifungal activity.

The revised text in the manuscript now reads (Lines 374–381):

“100 µL of bacterial suspension (cultured at 13°C and 200 r/min for 24–48 h) was inoculated onto Luria–Bertani (LB) agar plates (90 mm×18 mm). LB medium, composed mainly of tryptone (10 g/L), yeast extract (5 g/L), sodium chloride (10 g/L) and agar (15g/L), is a nutrient-rich medium commonly used for cultivating fast-growing bacteria with low nutritional requirements. We chose LB broth to culture the antagonistic bacterial strains because it allows for rapid accumulation of sufficient biomass under stable growth conditions, which is beneficial for the subsequent evaluation of antifungal activity.

We have added the following reference at lines 718–719: Bertani G. 1951. Studies on lysogenesis I: the mode of phage liberation by lysogenic *Escherichia coli*. J Bacteriol 62:293-300.

Lines 336- 343 and lines 346-352 appear to be describing the same experiment. It is unclear what was done as written.

Response: Thank you very much for your review and comments. We apologize for the previous ambiguity. We agree that these two experimental descriptions may appear similar on the surface, but they are in fact two independent experiments with different objectives and procedures:

Lines 370–383 describe a shared-space co-culture plate assay designed to screen for anti-*P. destructans* bacterial strains using a non-contact inhibition approach.

Lines 386–396 refer to a subsequent experiment that does not involve shared-space co-culture. In this case, the previously screened antagonistic bacterial strains and *P. destructans* were cultured separately for subsequent GC-MS analysis, with the primary goal of identifying and verifying the inhibitory effects of specific volatile organic compounds.

We have revised Lines 386–396 in the Methods section to clarify the independent plate culture experiment.

“In this subsequent experiment, the previously screened antagonistic bacterial strains and *P. destructans* were cultured separately, without shared-space co-culture. The antagonistic strains were inoculated in LB broth and cultured at 13°C and 200 r/min shaking for 24–48 h to prepare bacterial suspensions. Next, 100 µL of the antagonistic strain suspension was inoculated onto LB agar plates, while uninoculated LB agar plates served as controls. At the same time, 100 µL of *P. destructans* conidia suspension (2×10^6 conidia/mL) was evenly spread onto SDA agar plates, with uninoculated SDA plates as controls. All plates were incubated at 13°C for 14 days.

The purpose of this experiment was to prepare for as chromatography-mass spectrometry at identifying volatile organic compounds produced by anti-*P. destructans* bacterial strains.”

Paragraph 371- this is a contact-dependent experiment, so it is not necessarily testing VOCs.

Response: We thank the reviewer for this comment. We apologize for the lack of clarity in our original description. In this experiment, *P. destructans* was cultured on the SDA medium at the bottom of the Petri dish, while sterile antibiotic susceptibility discs loaded with the tested compounds were affixed to the inner surface of the Petri dish lid. Therefore, the discs and the fungal culture did not make direct contact. The inhibitory effect was mediated exclusively via volatilization and diffusion of the VOCs. We have revised the Methods section (lines 420–426) to clarify this point.

Revised text: To verify the inhibitory effects of VOCs on *P. destructans* growth, 100 μL of *P. destructans* conidia suspension (2×10^6 conidia/mL) was evenly spread onto the SDA agar surface at the bottom of a Petri dish (90 mm \times 18 mm). Sterile antibiotic susceptibility discs loaded with either 10 μL or 100 μL of one of the following compounds (2-Undecanone, 2-Tridecanone, Benzaldehyde, 1-Undecene, 2-Nonanone, Thujone, 2,5-Dimethylpyrazine, or α -Pinene) were affixed to the inner surface of the Petri dish lid (i.e., opposite the SDA agar surface).

Whether Pd is inhibited or not depends on the media type it is grown on, agar salt content, and agar pH. This was demonstrated in Vanderwolf et al. 2021. Skin fungal assemblages of bats vary based on susceptibility to white-nose syndrome. The ISME Journal. The fact that only 1 agar type and 1 temperature was chosen to conduct inhibition assays is a caveat of the study. It is unclear whether in vitro results apply in vivo.

Response: We thank the reviewer for this important comment. We acknowledge that the inhibition of *Pseudogymnoascus destructans* can be influenced by multiple factors including agar type, salt content, and pH, as demonstrated by Vanderwolf et al. (2021). In this study, we selected a single agar type (SDA) and a temperature of 13°C to simulate environmental conditions of caves in winter and to maintain consistency with previous non-contact antagonistic bacterial screening assays (Li et al., 2021).

We agree that this represents a limitation of the study. To address this, we have added the following statement to the Discussion section (lines 295–300):

One limitation of this study is that all inhibition assays were conducted using a single agar type (SDA) and a single incubation temperature (13°C). It is known that the growth and inhibition of *P.destructans* can be influenced by media composition, agar salt content, and pH (Vanderwolf et al. 2021). Therefore, we plan to conduct further studies under different media compositions, pH levels, and other culture conditions to validate the effectiveness of these inhibitory effects.

We have added the following reference at lines 704–706: Vanderwolf KJ, Campbell LJ, Goldberg TL, Blehert DS, Lorch JM. 2021. Skin fungal assemblages of bats vary based on susceptibility to white-nose syndrome. ISME J

Re: Spectrum01241-25R2 (Screening microbial inhibitors of *Pseudogymnoascus destructans* in northern China)

Dear Dr. Zhongle Li:

Your manuscript has been accepted, and I am forwarding it to the ASM production staff for publication. Your paper will first be checked to make sure all elements meet the technical requirements. ASM staff will contact you if anything needs to be revised before copyediting and production can begin. Otherwise, you will be notified when your proofs are ready to be viewed.

Sincerely,
Renato Kovacs
Editor
Microbiology Spectrum

Reviewer #2 (Comments for the Author):

I am satisfied the authors have adequately addressed my comments. I appreciate the amount of work they put in!